# Generalization Error Rates in Kernel Regression: The Crossover from the Noiseless to Noisy Regime

**Hugo Cui**
SPOC, EPFL, Lausanne
`hugo.cui@epfl.ch`

**Bruno Loureiro**
IdePHICS, EPFL, Lausanne

**Florent Krzakala**
IdePHICS, EPFL, Lausanne

**Lenka Zdeborová**
SPOC, EPFL, Lausanne

## Abstract

In this manuscript we consider Kernel Ridge Regression (KRR) under the Gaussian design. Exponents for the decay of the excess generalization error of KRR have been reported in various works under the assumption of power-law decay of eigenvalues of the features co-variance. These decays were, however, provided for sizeably different setups, namely in the noiseless case with constant regularization and in the noisy optimally regularized case. Intermediary settings have been left substantially uncharted. In this work, we unify and extend this line of work, providing characterization of all regimes and excess error decay rates that can be observed in terms of the interplay of noise and regularization. In particular, we show the existence of a transition in the noisy setting between the noiseless exponents to its noisy values as the sample complexity is increased. Finally, we illustrate how this crossover can also be observed on real data sets.

## 1 Introduction

Kernel methods are among the most popular models in machine learning. Despite their relative simplicity, they define a powerful framework in which non-linear features can be exploited without leaving the realm of convex optimisation. Kernel methods in machine learning have a long and rich literature dating back to the 60s [1, 2], but have recently made it back to the spotlight as a proxy for studying neural networks in different regimes, e.g. the infinite width limit [3–6] and the lazy regime of training [7]. Despite being defined in terms of a non-parametric optimisation problem, kernel methods can be mathematically understood as a standard parametric linear problem in a (possibly infinite) Hilbert space spanned by the kernel eigenvectors (a.k.a *features*). This dual picture fully characterizes the asymptotic performance of kernels in terms of a trade-off between two key quantities: the relative decay of the eigenvalues of the kernel (a.k.a. its *capacity*) and the coefficients of the target function when expressed in feature space (a.k.a. the *source*). Indeed, a sizeable body of work has been devoted to understanding the decay rates of the excess error as a function of these two relative decays, and investigated whether these rates are attained by algorithms such as stochastic gradient descent [8, 9].

Rigorous optimal rates for the excess generalization error in kernel ridge regression and are well-known since the seminal works of [10, 11]. However, recent interesting works [12, 13] surprisingly reported very different - and actually better - rates supported by numerical evidences. These papers appeared to either not comment on this discrepancy [13], or to attribute this apparent contradiction to a difference between typical and worse-case analysis [12]. As we shall see, the key difference between these works stems instead from the fact that most of classical works considered *noisy* data and fine-tuned regularization, while [12, 13] focused on noiseless data sets. This observation raises a number of questions: is there a connection between both sets of exponents? Are Gaussian design

35th Conference on Neural Information Processing Systems (NeurIPS 2021).

exponents actually different from worst-case ones? What about intermediary setups (for instance noisy labels with generic regularization, noiseless labels with varying regularization) and regimes (intermediary sample complexities)? How does infinitesimal noise differ from no noise at all?

**Main contributions —** In this manuscript, we answer all the above questions, and redeem the apparent contradiction by reconsidering the Gaussian design analysis. We provide a unifying picture of the decay rates for the excess generalization error, along a more exhaustive characterization of the regimes in which each is observed, evidencing the interplay of the role of regularization, noise and sample complexity. We show in particular that typical-case analysis with a Gaussian design is actually in perfect agreement with the statistical learning worst-case data-agnostic approach. We also show how the optimal excess error decay can transition from the recently reported noiseless value to its well known noisy value as the number of samples is increased. We illustrate this crossover from the *noiseless* regime to the *noisy* regime also in a variety of KRR experiments on real data.

**Related work —** The analysis of decay rates for kernel methods and ridge regression is a classical topic in statistical learning theory [10, 11, 14, 15]. In this classical setting, decay exponents for optimally regularized *noisy* linear regression on features with power-law co-variance spectrum have been provided. Interestingly, it has been shown that such optimal rates can be obtained in practice by stochastic gradient descent, without explicit regularization, with single-pass [16, 17] or multi-pass [8] algorithms, as well as by randomized algorithms [18]. Closed-form bounds for the prediction error have been provided in a number of worst-case analyses [18, 19]. We show how the decay rates given in the present paper can also be alternatively deduced therefrom in Appendix E.

The recent line of work on the noiseless setting includes contributions from statistical learning theory [9, 18] and statistical physics [12, 13]. This much more recent second line of work proved decay rates for a given, constant regularization. An example of noise-induced crossover is furthermore mentioned in [9]. The interplay between noisy and noiseless regimes has also been investigated in the related Gaussian Process literature [20].

The study of ridge regression with Gaussian design is also a classical topic. Ref. [21] considered a model in which the covariates are isotropic Gaussian in $\mathbb{R}^p$, and computed the exact asymptotic generalization error in the high-dimensional asymptotic regime $p, n \to \infty$ with dimension-to-sample-complexity ratio $p/n$ fixed. This result was generalised to arbitrary co-variances [22, 23] using fundamental results from random matrix theory [24]. Non-asymptotic rates of convergence for a related problems were given in [25]. Previous results also existed in the statistical physics literature, e.g. [26–29]. Gaussian models for regression have seen a surge of popularity recently, and have been used in particular to study over-parametrization and the double-descent phenomenon, e.g. in [30–43].

## 2 Setting

Consider a data set $\mathcal{D} = \{x^\mu, y^\mu\}_{\mu=1}^n$ with $n$ independent samples from a probability measure $\rho$ on $\mathcal{X} \times \mathcal{Y}$, where $\mathcal{X} \subset \mathbb{R}^d$ is the input and $\mathcal{Y} \subset \mathbb{R}$ the response space. Let $K$ be a kernel and $\mathcal{H}$ denote its associated reproducing kernel Hilbert space (RKHS). *Kernel ridge regression* (KRR) corresponds to the following non-parametric minimisation problem:

$$\min_{f \in \mathcal{H}} \frac{1}{n} \sum_{\mu=1}^n (f(x^\mu) - y^\mu)^2 + \lambda ||f||_{\mathcal{H}}^2. \tag{1}$$

where $|| \cdot ||_{\mathcal{H}}$ is the norm associated with the scalar product in $\mathcal{H}$, and $\lambda \geq 0$ is the regularisation. The convenience of KRR is that it admits a dual representation in terms of a standard parametric problem. Indeed, the kernel $K$ can be diagonalized in an orthonormal basis $\{\phi_k\}_{k=1}^\infty$ of $L^2(\mathcal{X})$:

$$\int_{\mathcal{X}} \rho_x(\mathrm{d}x') K(x, x') \phi_k(x') = \eta_k \phi_k(x) \tag{2}$$

where $\{\eta_k\}_{k=1}^\infty$ are the corresponding (non-negative) kernel eigenvalues and $\rho_x$ is the marginal distribution over $\mathcal{X}$. It is convenient to define the re-scaled basis of *kernel features* $\psi_k(x) = \sqrt{\eta_k}\phi_k(x)$ and to work in matrix notation in feature space: define $\phi(x) \equiv \{\phi_k(x)\}_{k=1}^p$ (with $p$ possibly infinite)

$$\psi(x) = \Sigma^{\frac{1}{2}}\phi(x) \qquad \mathbb{E}_{x \sim \rho_x}\left[\phi(x)\phi(x)^\top\right] = \mathbb{1}_p, \qquad \mathbb{E}_{x' \sim \rho_x}\left[K(x, x')\phi(x')\right] = \Sigma\phi(x), \tag{3}$$

where $\Sigma \equiv \mathbb{E}_{x \sim \rho_x} \left[ \psi(x)\psi(x)^\top \right] = \mathrm{diag}(\eta_1, \eta_2, ..., \eta_p)$ is the features co-variance (a diagonal operator in feature space). With this notation, we can rewrite eq. (4) in feature space as a standard parametric problem for the following empirical risk:

$$\hat{\mathcal{R}}_n(w) = \frac{1}{n} \sum_{\mu=1}^{n} \left( w^\top \psi(x^\mu) - y^\mu \right)^2 + \lambda \, w^\top w. \tag{4}$$

Our main results concern the typical averaged performance of the KRR estimator, as measured by the typical prediction (out-of-sample) error

$$\epsilon_g = \mathbb{E}_{\mathcal{D}} \mathbb{E}_{(x,y) \sim \rho} (\hat{f}(x) - y)^2 , \tag{5}$$

where the first average is over the data $\mathcal{D} = \{x^\mu, y^\mu\}$ and the second over a fresh sample $(x, y) \sim \rho$.

In what follows we assume the labels $y^\mu \in \mathcal{Y}$ were generated, up to an independent additive Gaussian noise with variance $\sigma^2$, by a target function $f^\star$ (not necessarily belonging to $\mathcal{H}$):

$$y^\mu \overset{d}{=} f^\star(x^\mu) + \sigma\mathcal{N}(0,1), \tag{6}$$

and we denote by $\theta^\star$ the coefficients of the target function in the features basis $f^\star(x) = \psi(x)^\top \theta^\star$. As we will characterize below, whether the target function $f^\star$ belongs or not to $\mathcal{H}$ depends on the relative decay coefficients $\theta^\star$ with respect to the eigenvalues of the kernel. We often refer to $\theta^\star$ as the *teacher*. While the present results and discussion are provided for additive gaussian noise for simplicity, our method are not restricted to this particular form of noise, and a more complete extension of the results for other noise settings is left for future work. We are then interested in the evolution of the *excess error* $\epsilon_g - \sigma^2$ as the number of samples $n$ is increased.

**Capacity and source coefficients —** Motivated by the discussion above, we focus on ridge regression in an infinite dimensional ($p \to \infty$) space $\mathcal{H}$ with Gaussian design $u^\mu \overset{\text{def}}{=} \psi(x^\mu) \overset{d}{=} \mathcal{N}(0, \Sigma)$ with (without loss of generality) diagonal co-variance $\Sigma = \mathrm{diag}(\eta_1, \eta_2, ...)$. We expect however the results of this manuscript to be universal for a large class of distributions beyond the Gaussian one. In particular, we anticipate that the gaussianity assumption should be amenable to being relaxed to sub-gaussians [44] or even any concentrated distribution [45, 46].

Following the statistical learning terminology, we introduce two parameters $\alpha > 1, r \geq 0$, herefrom referred to as the *capacity* and *source* conditions [14], to parametrize the difficulty of the target function and the learning capacity of the kernel

$$\mathrm{tr}\, \Sigma^{\frac{1}{\alpha}} < \infty, \qquad\qquad ||\Sigma^{\frac{1}{2}-r}\theta^\star||_{\mathcal{H}} < \infty. \tag{7}$$

As in [9, 12, 13, 23], we consider the particular case where both the spectrum of $\Sigma$ and the teacher components $\theta_k^\star$ have exactly a power-law form satisfying some source/capacity conditions :

$$\eta_k = k^{-\alpha}, \qquad\qquad \theta_k^\star = k^{-\frac{1+\alpha(2r-1)}{2}} . \tag{8}$$

While this follows the standard naming convention, it is useful to translate these notations to the ones used in others works, and we give a dictionary in Appendix B, Table 2. For instance, $\alpha$ is $b$ and $r$ is $(a-1)/2b$ in [13], while the change of variable $\alpha = \alpha_S/d$ and $r = (\alpha_T/\alpha_S - d)/2$ allows to recover the notation from [12]. The power law ansatz (8) is empirically observed to be a rather good approximation for some real simple datasets and kernels, see Fig. 7 in Appendix C. The parameters $\alpha, r$ introduced in (8) control the complexity of the data and of the teacher respectively. A large $\alpha$ can be seen as characterizing an effectively low dimensional (and therefore easy to fit) data distribution. By the same token, a large $r$ signals a good alignment of the teacher with the important directions of the data co-variance, and therefore an a priori simple learning task. The regularization $\lambda$ is allowed to vary with $n$ according to a power-law $\lambda = n^{-\ell}$. This very general form allows us to encompass both the zero regularization case (corresponding to $\ell = \infty$) and the case where $\lambda = \lambda^\star$ is optimized, with some optimal decay rate $\ell^\star$. Note also that this power law form implies that $\lambda$ is assumed positive. While this is indeed the assumption of [10, 14] with which we intend to make contact, [37] have shown that the optimal $\lambda$ may in some settings may be negative. Some numerical experiments suggest that removing the positivity constraint on $\lambda$ while optimizing does not affect the results presented in this manuscript. A more detailed investigation is left to future work.

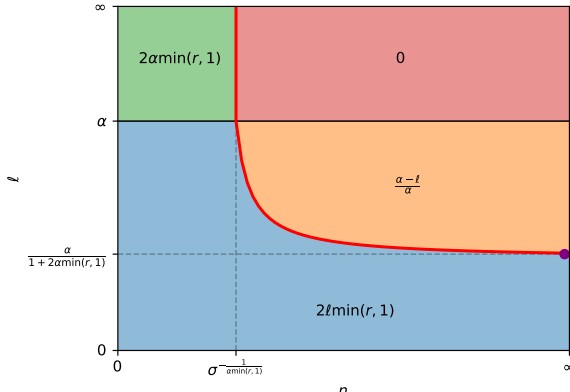

Figure 1: Different decays for the excess generalization error $\epsilon_g - \sigma^2$ for different values of $n$ and different decays $\ell$ of the regularization $\lambda \sim n^{-\ell}$, at given noise variance $\sigma$. The red solid line represents the noise-induced crossover line, separating the effectively noiseless regime (green and blue) on its left from the effectively noisy regime (red and orange) on its right. Any KRR experiment at fixed regularization decay $\ell$ (corresponding to drawing a horizontal line at ordinate $\ell$) crosses the crossover line if $\ell > \alpha/(1 + 2\alpha\min(r,1))$. The corresponding learning curve will accordingly exhibit a crossover from a fast decay (noiseless regime) to a slow decay (noisy regime).

## 3 Main results

Depending on the regularization decay strength $\ell$, capacity $\alpha$, source $r$ and noise variance $\sigma^2$, four regimes can be observed. The derivation of these decays from the asymptotic solution of the Gaussian design problem is sketched in Section 4 and detailed in Appendix A, and here we concentrate on the key results. The different observable decays for the excess error $\epsilon_g - \sigma^2$ are summarized in Fig. 1, and are given by:

- If $\ell \geq \alpha$ (weak regularization $\lambda = n^{-\ell}$),

$$\epsilon_g - \sigma^2 = \mathcal{O}\left(\max\left(\sigma^2, n^{-2\alpha\min(r,1)}\right)\right) . \tag{9}$$

The excess error transitions from a fast decay $2\alpha\min(r,1)$ (green region in Fig. 1 and green dashed line in Fig. 2) to a plateau (red region in Fig. 1 and red dashed line in Fig. 2) with no decay as $n$ increases. This corresponds to a crossover from the green region to the red region in the phase diagram Fig. 1.

- If $\ell \leq \alpha$ (strong regularization $\lambda = n^{-\ell}$),

$$\epsilon_g - \sigma^2 = \mathcal{O}\left(\max\left(\sigma^2, n^{1-2\ell\min(r,1)-\frac{\ell}{\alpha}}\right) n^{\frac{\ell-\alpha}{\alpha}}\right) . \tag{10}$$

The excess error transitions from a fast decay $2\ell\min(r,1)$ (blue region in Fig. 1) to a slower decay $(\alpha - \ell)/\alpha$ (orange region in Fig. 1) as $n$ is increased and the effect of the additive noise kicks in, see Fig. 3. The crossover disappears for too slow decays $l \leq \alpha/(1 + 2\alpha\min(r,1))$, as the regularization $\lambda$ is always sufficiently large to completely mitigate the effect of the noise. This corresponds to the max in (10) being realized by its second argument for all $n$.

Given these four different regimes as depicted in Fig. 1, one may wonder about the optimal learning solution when the regularization is fine tuned to its best value. To answer this question, we further define the *asymptotically optimal* regularization decay $\ell^\star$ as the value leading to fastest decay of the typical excess error $\epsilon_g - \sigma^2$. We find that two different optimal rates exist, depending on the quantity of data available.

- If $n \ll n_1^* \approx \sigma^{-\frac{1}{\alpha\min(r,1)}}$, any $\ell^\star \in (\alpha, \infty)$ yields excess error decay

$$\epsilon_g^\star - \sigma^2 \sim n^{-2\alpha\min(r,1)} . \tag{11}$$

- If $n \gg n_2^* \approx \sigma^{-\max\left(2, \frac{1}{\alpha\min(r,1)}\right)}$,

$$\epsilon_g^\star - \sigma^2 \sim n^{\frac{1}{1+2\alpha\min(r,1)}-1} , \qquad\qquad \lambda^\star \sim n^{-\frac{\alpha}{1+2\alpha\min(r,1)}} . \tag{12}$$

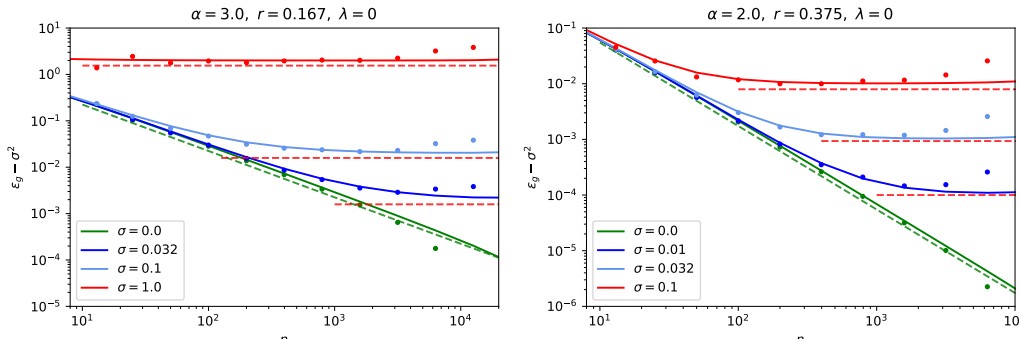

Figure 2: Kernel ridge regression on synthetic data sets with capacity $\alpha$ and source coefficient $r$, with no regularization $\lambda = 0$. Solid lines correspond to the theoretical prediction of eq. (14) using the `GCM` package associated with [25]. Points are simulations conducted using the python `scikit-learn` `KernelRidge` package [47], where the feature space dimension has been cut off to $p = 10^4$ for the simulations, and to $10^5$ for the theoretical curves. Dashed lines represent the slopes predicted by eq. (9), with the color (red and green) in correspondence to the regime from Fig. 1. The code used to generate this figure is available at https://github.com/IdePHICS/KernelRidgeCrossover.

The optimal decay for the excess error $\epsilon_g^\star - \sigma^2$ thus transitions from a fast decay $2\alpha\min(r, 1)$ when $n \ll n_1^*$ – corresponding to, effectively, the optimal rates expected in a "noiseless" situation – to a slower decay $2\alpha\min(r, 1)/(1 + 2\alpha\min(r, 1))$ when $n \gg n_2^*$ corresponding to the classical "noisy" optimal rate, depicted with the purple point in Fig. 1. This is illustrated in Fig. 4 where the two rates are observed in succession for the same data as the number of points is increased.

We can now finally clarify the apparent discrepancy in the recent literature discussed in the introduction. The exponent recently reported in [12, 13] actually corresponds to the "noiseless" regime. In contrast, the rate described in (12) is the classical result [10] for the non-saturated case $r < 1$ for generic data. We see here that the same rate is also achieved with Gaussian design, and that there are no differences between fixed and Gaussian design as long as the capacity and source condition are matching. We unveiled, however, the existence of two possible sets of optimal rate exponents depending on the number of data samples.

All setups (effectively non-regularized KRR (9), effectively regularized KRR (10) or optimally regularized KRR (11), (12)) can therefore exhibit a *crossover* from an effectively *noiseless* regime (green or blue in Fig. 1), to an effectively *noisy* regime (red, orange in Fig. 1) depending on the quantity of data available. We stress that while the noise is indeed present in the green and blue "noiseless" regimes, its presence is effectively not felt, and noiseless rates are observed. In fact, if the noise is small, one will not observed the classical noisy rates unless an astronomical amount of data is available. This can be intuitively understood as follows: for small sample size $n$, low-variance dimensions are used to overfit the noise, while the spiked subspace of large-variance dimensions is well fitted. In noiseless regions, the excess error is thus characterized by a fast decay. This phenomenon, where the noise variance is diluted over the dimensions of lesser importance, is connected to the *benign overfitting* discussed by [38] and [44]. Benign overfitting is possible due to the decaying structure of the co-variance spectrum (8). As more samples are accessed, further decrease of the excess error requires good generalization also over the low-variance subspace, and the overfitting of the noise results in a slower decay.

While our analysis is for the optimal full-batch learning, we note that a similar crossover in the case of SGD in the effectively non-regularized case (from green to red) has been discussed in [9, 48]. Note that the rates derived in these two works for SGD are slower than the green decay rate (9) for full-batch learning. It would be interesting to further explore how SGD can behave in the different regimes discussed here.

When $\lambda = \lambda_0 n^{-\ell}$ for a prefactor $\lambda_0$ that is allowed to be very small, a *regularization-induced* crossover, similar to the one reported in [13], can also be observed on top of the noise-induced crossover which is the focus of the present work. This setting is detailed in Appendix. D.

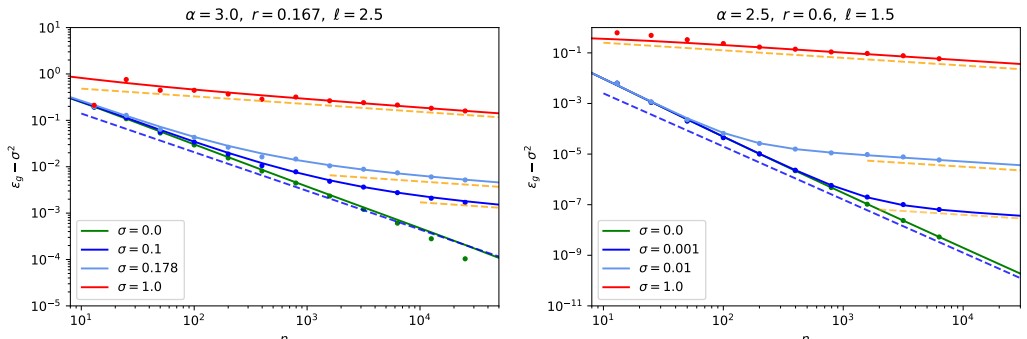

Figure 3: Kernel ridge regression on synthetic data sets with capacity $\alpha$ and source coefficient $r$, with regularization $\lambda = n^{-\ell}$. Solid lines correspond to the theoretical prediction of eq. (14) using the `GCM` package associated with [25]. Points are simulations conducted using the python `scikit-learn` `KernelRidge` package [47], where the feature space dimension has been cut off to $p = 10^4$ for the simulations, and to $10^5$ for the theoretical curves. Dashed lines represent the slopes predicted by eq. (10), with the color (blue and orange) in correspondence to the regime from Fig. 1. The code used to generate this figure is available at https://github.com/IdePHICS/KernelRidgeCrossover.

## 4 Sketch of the derivation

We provide in this section the main ideas underlying the derivation of the main results exposed in section 3 and summarized in Fig. 1. A more detailed discussion is presented in Appendix A.

**Closed-form solution for Gaussian design —** The starting point is to consider the closed-form, rigorous solution of the risk of ridge regression with Gaussian data of arbitrary co-variance in the high-dimensional asymptotic regime [23, 37, 39]. We shall use here the equivalent notations of [25], who have the advantage of having rigorous non-asymptotic rates guarantees. Within this framework, we shall sketch how the crossover phenomena (9) (10)(11) and (12), which are the main contribution of this paper, can be derived. With high-probability when $n, p$ are large the excess prediction error is expressed as

$$\epsilon_g - \sigma^2 = \rho - 2m^\star + q^\star, \tag{13}$$

with $\rho = \theta^{\star\top}\Sigma\theta^\star$, and $(m^\star, q^\star)$ are the unique fixed-points of the following self-consistent equations:

$$\begin{cases} \hat{V} = \frac{\frac{n}{p}}{1+V} \\ \hat{q} = \frac{n}{p}\frac{\rho+q-2m+\sigma^2}{(1+V)^2} \end{cases} , \quad \begin{cases} q = p\sum_{k=1}^{p}\frac{\hat{q}\eta_k^2+\theta_k^{\star 2}\eta_k^2\hat{m}^2}{(n\lambda+p\hat{V}\eta_k)^2} \\ m = p\hat{V}\sum_{k=1}^{p}\frac{\theta_k^{\star 2}\eta_k^2}{n\lambda+p\hat{V}\eta_k} \end{cases} , \quad \begin{cases} V = \frac{1}{p}\sum_{k=1}^{p}\frac{p\eta_k}{n\lambda+p\hat{V}\eta_k} \end{cases} . \tag{14}$$

We recall the reader that $\lambda > 0$ is the regularisation strength and $\{\eta_k\}_{k=1}^{p}$ are the kernel eigenvalues. The next step is thus to insert the power-law decay (8) for the eigenvalues into (14), and to take the limit $n, p \to \infty$. We note, however, that this last step is not completely justified rigorously. Indeed, [23] assumes $p/n = O(1)$ as $n, p \to \infty$ while here we first send $p \to \infty$ and then take the large $n$ limit, thus working effectively with $p/n \to 0$. While the non-asymptotic rates guarantees of [25] are reassuring in this respect, a finer control of the limit would be needed for a fully rigorous justification; perhaps using the tights non-asymptotic bounds from [44]. Nevertheless, we observed in our experiments that the agreement between theory and numerical simulations for the the excess prediction error (5) is perfect (see Figs. 2, 3 and 4). In the large $n$ limit, one can finally close the equation for the excess prediction error into

$$\epsilon_g - \sigma^2 = \frac{\sum_{k=1}^{\infty}\frac{k^{-1-2r\alpha}}{(1+nz^{-1}k^{-\alpha})^2}}{1 - \frac{n}{z^2}\sum_{k=1}^{\infty}\frac{k^{-2\alpha}}{(1+nz^{-1}k^{-\alpha})^2}} + \sigma^2\frac{\frac{n}{z^2}\sum_{k=1}^{\infty}\frac{k^{-2\alpha}}{(1+nz^{-1}k^{-\alpha})^2}}{1 - \frac{n}{z^2}\sum_{k=1}^{\infty}\frac{k^{-2\alpha}}{(1+nz^{-1}k^{-\alpha})^2}}. \tag{15}$$

with $z$ being a solution of

$$z \approx n\lambda + \left(\frac{z}{n}\right)^{1-\frac{1}{\alpha}}\int_{\left(\frac{z}{n}\right)^{1/\alpha}}^{\infty}\frac{\mathrm{d}x}{1+x^\alpha}. \tag{16}$$

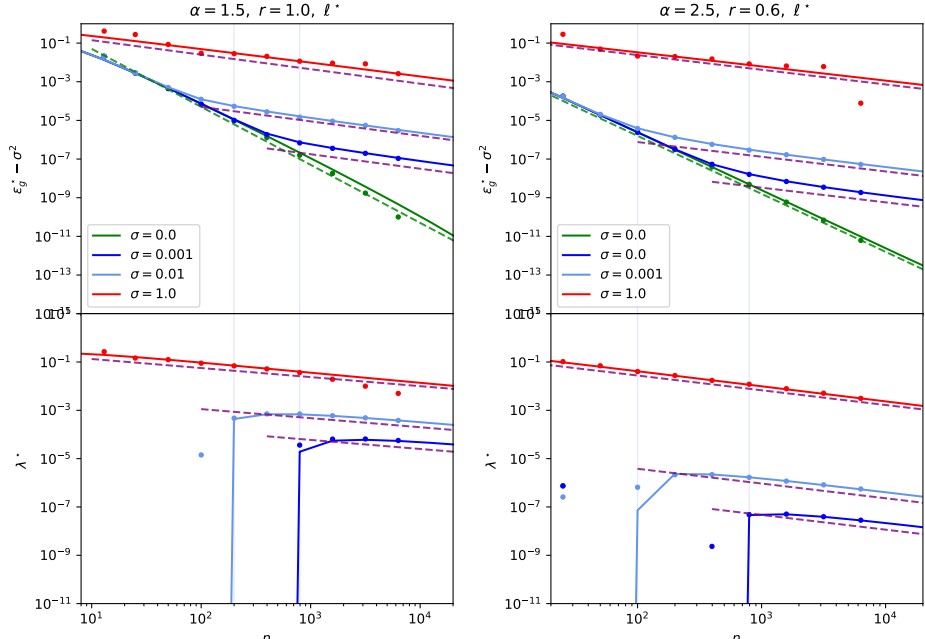

Figure 4: Kernel ridge regression on synthetic data sets with capacity $\alpha$ and source coefficient $r$. The regularization $\lambda$ is chosen as the one minimizing the theoretical prediction for the excess generalization error, deduced from eq. (14) using the `GCM` package associated with [25]. Solid lines correspond to the theoretical prediction of eq. (14). Points are simulations conducted with the python `scikit-learn KernelRidge` package [47], where the feature space dimension has been cut off to $p = 10^4$ for the simulations, and to $10^5$ for the theoretical curves. In simulations, the best $\lambda^\star$ was determined using python `scikit-learn GridSearchCV` cross validation package [47]. Note that because cross validation is not adapted to small training sets, a few discrepancies are observed for smaller $n$. Dashed lines represent the slopes predicted by theory, with the colors in correspondence to the regimes in Fig. 1, purple for the purple point in Fig. 1. Top: excess error. Bottom: optimal $\lambda^\star$. Note the noiseless case has $\lambda^* = 0$. The code used to generate this figure is available at https://github.com/IdePHICS/KernelRidgeCrossover.

The detailed derivation is provided in Appendix A. We note that this equation was observed with heuristic arguments from statistical physics (using the non-rigorous cavity method) in [49].

The different regimes of excess generalization error rates discussed in Section 3 are derived from this self-consistent equation. Note that the excess error (15) decomposes over a sum of two contributions, respectively accounting for the sample variance and the noise-induced variance. In contrast to a typical bias-variance decomposition, the effect of the bias introduced in the task for non-vanishing $\lambda$ is subsumed in both terms.

**Derivation of the four regimes —** If the second term in (16) dominates, then $z \sim n^{1-\alpha}$, which is self consistent if $\ell \geq \alpha$. This is the *effectively non-regularized regime*, where the regularization $\lambda$ is not sensed, and corresponds to the green and red regimes in the phase diagram in Fig. 1. This scaling of $z$ can then be used to estimate the asymptotic behaviour of the sample and noise induced variance in the decomposition on the excess error (15), yielding

$$\epsilon_g - \sigma^2 = \mathcal{O}(n^{-2\alpha\min(r,1)}) + \sigma^2\mathcal{O}(1), \tag{17}$$

which can be rewritten more compactly as (9). Therefore, for small sample sizes the sample variance drives the decay of the excess prediction error, while for larger samples sizes the noise variance dominates and causes the error to plateau. The crossover happens when both variance terms in (17) are balanced, around

$$n \sim \sigma^{-\frac{1}{\alpha\min(r,1)}}, \tag{18}$$

which corresponds to the vertical part of the crossover line in Fig. 1.

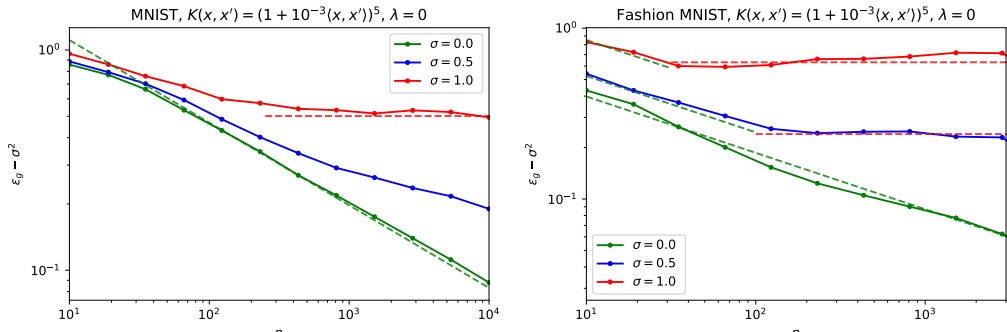

Figure 5: Excess error for MNIST odd versus even (above) and Fashion MNIST t-shirt versus coat (below) with labels corrupted by noise of variance $\sigma^2$. The kernel used is indicated in the title. Solid lines with points come from numerical experiments with zero regularization. Dashed lines are the slopes $-2\alpha r$ (as $r < 1$) or $0$, predicted by the theory from the empirical values of $\alpha, r$ measured from the Gram matrix spectrum and the teacher for each data set, see Table 1. Colors of the dashed lines (green & red) indicate the regimes in Fig. 1. The code used to generate this figure is available at https://github.com/IdePHICS/KernelRidgeCrossover.

If the first term $n\lambda$ dominates in (16), then $z \sim n\lambda$, which is consistent provided that $\ell < \alpha$. This is the *effectively regularized regime* (blue, orange regions in Fig. 1). The two variances in (15) are found to asymptotically behave like

$$\epsilon_g - \sigma^2 = \mathcal{O}(n^{-2\ell\min(r,1)}) + \sigma^2\mathcal{O}(n^{\frac{\ell-\alpha}{\alpha}}), \tag{19}$$

which can be rewritten more compactly as (10). If the decay of the noise variance term $(\alpha - \ell)/\alpha$ is faster than the $2\ell\min(r, 1)$ decay of the sample variance term, then the latter always dominates and no crossover is observed. This is the case for $\ell < \alpha/(1 + 2\alpha\min(r, 1))$. If on the contrary the decay of the noise variance term is the slowest, then this term dominates at larger $n$, with a crossover when both terms in (19) are balanced, around

$$n \sim \sigma^{\frac{2}{1-\frac{\ell}{\alpha}(1+2\alpha\min(r,1))}} \tag{20}$$

Eqs. (17) and (19) are respectively equivalent to (9) and (10), and completely define the four regimes observable in Fig. 1. Equations (20) and (18) give the expression for the crossover line in Fig. 1.

**Asymptotically optimal regularization —** Determining the asymptotically optimal $\ell^\star$ is a matter of finding the $\ell$ leading to fastest excess error decay. We focus on the far left part and the far right part of the phase diagram Fig. 1.

In the $n \gg n_2^\star \approx \sigma^{-\max\left(2, \frac{1}{\alpha\min(r,1)}\right)}$ limit where the crossover line confounds itself with its $\ell = \alpha/(1 + 2\alpha\min(r, 1))$ asymptot, this is tantamount to solving the maximization problem

$$\ell^\star = \underset{\ell}{\mathrm{argmax}} \left( 2\ell\min(r, 1)\mathbb{1}_{0<\ell<\frac{\alpha}{(1+2\alpha\min(r,1))}} + \frac{\alpha-\ell}{\alpha}\mathbb{1}_{\frac{\alpha}{(1+2\alpha\min(r,1))}<\ell<\alpha} + 0 \times \mathbb{1}_{\alpha<\ell} \right) \tag{21}$$

which admits as solution (12). In the $n \ll n_1^\star \approx \sigma^{-\frac{1}{\alpha\min(r,1)}}$ range, the maximization of the excess error decay reads

$$\ell^\star = \underset{\ell}{\mathrm{argmax}} \left( 2\ell\min(r, 1)\mathbb{1}_{0<\ell<\alpha} + 2\alpha\min(r, 1)\mathbb{1}_{\alpha<\ell} \right), \tag{22}$$

and admits as solution (11).

## 5 Illustration on simple real data sets

In this section we show that the derived decay rates can indeed be observed in real data sets with labels artificially corrupted by additive Gaussian noise. For real data, the decay model in eq. (8) is idealized, and in practice there is no firm reason to expect a power-law decay. However, we do find

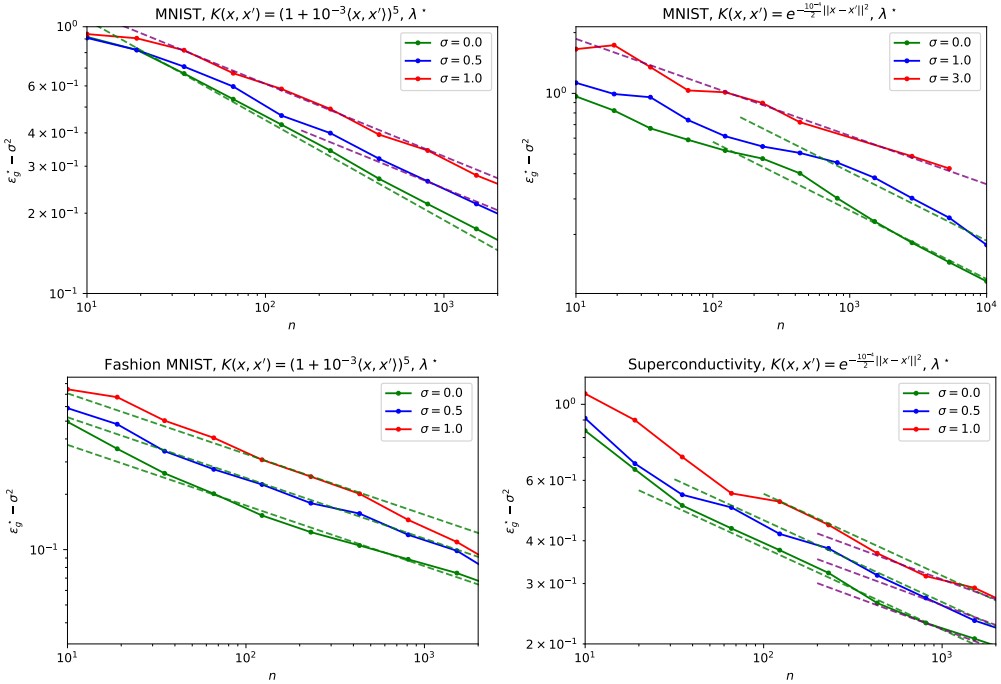

Figure 6: Excess error for MNIST odd versus even, and Fashion MNIST t-shirt versus coat, and the critical temperature regression. The kernel used is indicated in the title. Solid lines with dots come from numerical experiments with the regularization optimized using the python `scikit-learn GridSearchCV` package [47]. Dashed lines are the slopes predicted by the theory, from the empirical values of $\alpha, r$ measured from the Gram matrix spectrum and the teacher for each data set, see Table 1. Colors of the dashed lines indicate the regime in Fig. 1. The code used to generate this figure is available at https://github.com/IdePHICS/KernelRidgeCrossover.

that for some of the data sets and kernels we investigated, the power law fit is reasonable and can be used to estimate the exponents $\alpha$ and $r$, see Appendix C for details. For those cases, we compare the theoretically predicted exponents, eqs. (9), (10), (11) and (12) with the empirically measured learning curve, and obtain a very good agreement. We stress that the decay rates are not obtained by fitting the learning curves, but rather by fitting the exponents $\alpha$ and $r$ from the data. We also observe the crossover from the noiseless (blue, green in Fig. 1) to the noisy (orange, red in Fig. 1) regime given by the theory. Here we illustrate this with the learning curves for the following three data sets:

• MNIST even versus odd, a data set of $7 \times 10^4$ $28 \times 28$ images of handwritten digits. Even (odd) digits were assigned label $y = 1 + \sigma\mathcal{N}(0,1)$ ($y = -1 + \sigma\mathcal{N}(0,1)$).
• Fashion MNIST t-shirts versus coats, a data set of $14702$ $28 \times 28$ images of clothes from an online shopping platform [50]. T-shirts (coats) were assigned label $y = 1 + \sigma\mathcal{N}(0,1)$ ($y = -1 + \sigma\mathcal{N}(0,1)$).
• Superconductivity [51], a data set of 81 attributes of 21263 superconducting materials. The target $y^\mu$ corresponds to the critical temperature of the material, corrupted by additive Gaussian noise.

Learning curves are illustrated for a radial basis function (RBF) kernel $K(x,x') = e^{-\frac{\gamma}{2}||x-x'||^2}$ with parameter $\gamma = 10^{-4}$ and a degree 5 polynomial kernel $K(x,x') = (1 + \gamma\langle x, x'\rangle)^5$ with parameter $\gamma = 10^{-3}$. In Fig. 5 the regularization $\lambda$ was set to 0, while in Fig. 6 $\lambda$ was optimized for each sample size $n$ using the python `scikit-learn GridSearchCV` package [47]. KRR was carried out using the `scikit-learn KernelRidge` package [47]. The values of $\alpha, r$ were independently measured (see Appendix C) for each data set, and the estimated values summarized in Table 1. From these values the theoretical decays (9), (11) and (12) were computed, and compared with the simulations with very good agreement. Since for real data the power-law form (8) does not exactly hold (see Fig. 7 in the appendix), the estimates for $\alpha, r$ slightly vary depending on how the power-law is fitted. The precise procedure employed is described in Appendix C. Overall this variability does not hurt the good agreement with the simulated learning curves in Fig. 5 and 6.

| Dataset | Kernel | $\alpha$ | $r$ |
|---|---|---|---|
| Fashion MNIST | $K(x,x') = (1 + 10^{-3}\langle x, x'\rangle)^5$ | 1.3 | 0.13 |
| MNIST | $K(x,x') = (1 + 10^{-3}\langle x, x'\rangle)^5$ | 1.2 | 0.15 |
| MNIST | $K(x,x') = \exp(-10^{-4}||x - x'||^2/2)$ | 1.65 | 0.097 |
| Superconductivity | $K(x,x') = \exp(-10^{-4}||x - x'||^2/2)$ | 2.7 | 0.046 |

Table 1: Values of the source and capacity coefficients (7) as estimated from the data sets. The details on the estimation procedure can be found in Appendix C.

When $\lambda = 0$ (Fig. 5) the characteristic plateau for large label noises is observed for both MNIST & Fashion MNIST. For polynomial kernel regression on Fashion MNIST (Fig. 5 right), the crossover between noiseless (slope $-2\alpha r$ as $r < 1$) and noisy (slope 0) regimes is apparent on the same learning curve at noise levels $\sigma = 0.5, 1$. For MNIST, the $\sigma = 0$ ($\sigma = 1$) curve is in the noiseless (noisy) regime for larger $n$, while at intermediary noise $\sigma = 0.5$, and small $n$ for $\sigma = 1$, the curve is in the crossover regime between noiseless and noisy, consequently displaying in-between decay. Our results for the decays for $\sigma = 0$ agree with simulations for RBF regression on MNIST provided in [12].

For optimal regularization $\lambda = \lambda^\star$ (Fig. 6), as the measured $r < 1$ we have exponents $-2r\alpha$ for the noiseless regime and $-2r\alpha/(1 + 2r\alpha)$ for noisy. Since the measured value of $2r\alpha$ is rather small the difference between the two rates is less prominent. Nevertheless, it seems that in our experiments the noisy regime is observed for polynomial and RBF kernels on MNIST and $\sigma = 0.5, 1$. For Superconductivity, the green and purple decay have close values and it is difficult to clearly identify the regime. For Fashion MNIST only the noiseless rate is observable in the considered noise range and sample range.

**Conclusion —** To conclude, we unify hitherto disparate lines of work, and give a comprehensive study of observable regimes, along the associated decay rates for the excess error, for kernel ridge regression with features having power-law co-variance spectrum. We show that the effect of the noise only kicks in at larger sample complexity, meaning, in particular, that the KRR transitions from a *noiseless* regime with fast error decay to a *noisy* regime with slower decay. This crossover is shown to happen for zero, decaying and optimized regularization, and is observed on a variety of real data sets corrupted with label noise.

Due to the theoretical nature of this paper, no negative societal impact of this work is anticipated.

**Acknowledgments —** We warmly thank Loucas Pillaud-Vivien for his insights, and for his help in navigating the literature on kernel ridge regression. We also thank Volkan Cevher, Cedric Gerbelot and Matthieu Wyart for useful discussions. We acknowledge funding from the ERC under the European Union's Horizon 2020 Research and Innovation Programme Grant Agreement 714608-SMiLe, and from the French National Research Agency grants ANR-17-CE23-0023-01 PAIL.

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
