# Supplementary material to Generalization Error Rates in Kernel Ridge Regression : The Crossover from the Noiseless to Noisy Regime

## A Derivation of the decays

### A.1 Equations for Gaussian design

In this Appendix we discuss the derivation of eqs. (13), (14) describing the excess prediction error for the ridge regression problem with generic covariance matrix. Exact asymptotic formulas for the excess prediction error of least-squares and ridge regression are a classic result in high-dimensional statistics, and have been derived in many different works [23, 32, 52, 53]. In this manuscript, we follow the presentation given in [25], which is particularly adapted to our derivation and has the advantage to hold rigorously at large but finite number of samples $n$ and features $p$.

We start by reviewing the formulas in [25]. Consider the ridge regression problem on $n$ independent $p$-dimensional samples $\{u^\mu, y^\mu\}_{\mu=1}^n$, defined by a minimisation of the following empirical risk:

$$\hat{\mathcal{R}}_n(w) = \sum_{\mu=1}^n \left( \frac{w \cdot u^\mu}{\sqrt{p}} - y^\mu \right)^2 + \lambda ||w||_2^2. \tag{23}$$

Assume a Gaussian design $u^\mu \overset{d}{=} \mathcal{N}(0, \Sigma)$ with diagonal covariance $\Sigma = \mathrm{diag}(\eta_1, ..., \eta_p)$ and labels $y^\mu$ generated from a teacher/target/oracle $\theta^\star \in \mathbb{R}^p$:

$$y^\mu = \frac{\theta^\star \cdot u^\mu}{\sqrt{p}} + \sigma \mathcal{N}(0, 1). \tag{24}$$

Under the assumptions

**(A1)** $n \gg 1, p \gg 1, n/p = \mathcal{O}(1)$,

**(A2)** $0 < ||\theta^\star||^2/p < \infty$,

there exists constants $C, c, c' > 0$ such that for all $0 < \epsilon < c'$,

$$\mathbb{P}\left( |\epsilon_g - \sigma^2 - (\rho - 2m^\star + q^\star)| > \epsilon \right) < \frac{C}{\epsilon} e^{-cn\epsilon^2}. \tag{25}$$

where $\rho = \theta^\star \cdot \Sigma \cdot \theta^\star/p$, and $(m^\star, q^\star)$ are fixed-points of the following *self-consistent equations*

$$\begin{cases} \hat{V} = \hat{m} = \frac{\frac{n}{p}}{1+V} \\ \hat{q} = \frac{n}{p} \frac{\rho+q-2m+\sigma^2}{(1+V)^2} \end{cases}, \qquad \begin{cases} V = \frac{1}{p} \sum_{k=1}^p \frac{\eta_k}{\lambda+\hat{V}\eta_k} \\ q = \frac{1}{p} \sum_{k=1}^p \frac{\hat{q}\eta_k^2+\theta_k^{\star 2}\eta_k^2\hat{m}^2}{(\lambda+\hat{V}\eta_k)^2} \\ m = \frac{\hat{m}}{p} \sum_{k=1}^p \frac{\theta_k^{\star 2}\eta_k^2}{\lambda+\hat{V}\eta_k} \end{cases}. \tag{26}$$

Note that the risk considered in eq. (23) slightly differs from eq. (4) by: a) a $1/n$ factor multiplying the sum, b) additional $\sqrt{p}$ scalings and c) the fact that it is written for finite $p$. Accounting for these differences, we can rewrite Theorem 1 of [25] in our setting as:

$$\epsilon_g - \sigma^2 = \lim_{p \to \infty} (\rho - 2m^\star + q^\star), \tag{27}$$

with $\rho = \theta^{\star\top}\Sigma\theta^\star$, and $(m^\star, q^\star)$ fixed-points of

$$\begin{cases} \hat{V} = \frac{\frac{n}{p}}{1+V} \\ \hat{q} = \frac{n}{p} \frac{\rho+q-2m+\sigma^2}{(1+V)^2} \end{cases}, \qquad \begin{cases} V = \frac{1}{p} \sum_{k=1}^p \frac{p\eta_k}{n\lambda+p\hat{V}\eta_k} \\ q = p \sum_{k=1}^p \frac{\hat{q}\eta_k^2+\theta_k^{\star 2}\eta_k^2\hat{m}^2}{(n\lambda+p\hat{V}\eta_k)^2} \\ m = p\hat{V} \sum_{k=1}^p \frac{\theta_k^{\star 2}\eta_k^2}{n\lambda+p\hat{V}\eta_k} \end{cases}. \tag{28}$$

Note, however, that rescaling from (26) to (28), sending $p \to \infty$ while keeping $n$ finitely large, and further allowing $\lambda$ to scale with $n$ all break the initial assumptions of Theorem 1 [25], thereby losing the control in eq. (25). Therefore, strictly speaking the results derived hereafter are not rigorous, and we assume that the typical excess error can still be computed from eq. (27). In fact, this is well-justified by comparing the results obtained from extrapolating the theory with finite instance simulation, e.g. Figs. 2, 3, and 4.

## A.2 Self-consistent equations for the excess prediction error

Defining $z = \frac{n^2}{p}\frac{\lambda}{\hat{V}}$, the equations (28) allow to write

$$z = n\lambda + \frac{z}{n}\sum_{k=1}^{p}\frac{\eta_k}{\frac{z}{n}+\eta_k}. \tag{29}$$

An expression for the excess error $\epsilon_g - \sigma^2$ can be obtained combining (27) with (28):

$$\epsilon_g - \sigma^2 \underset{(a)}{=} \lim_{p\to\infty}\frac{1}{p}\sum_{k=1}^{p}\left[\theta_k^{\star 2}p\eta_k + \frac{\hat{q}p^2\eta_k^2 + \theta_k^{\star 2}p^2\eta_k^2\hat{m}^2}{(n\lambda + \hat{V}p\eta_k)^2} - \frac{2\hat{m}\theta_k^{\star 2}p^2\eta_k^2}{n\lambda + \hat{V}p\eta_k}\right]$$

$$\underset{(b)}{=} \lim_{p\to\infty}\sum_{k=1}^{p}\frac{\theta_k^{\star 2}\eta_k\left(n\lambda + \hat{V}p\eta_k\right)^2 + \frac{p^2}{n}\eta_k^2\hat{V}^2\epsilon_g + \hat{V}^2\theta_k^{\star 2}p\eta_k^2 - 2\theta_k^{\star 2}\hat{V}p\eta_k^2\left(n\lambda + \hat{V}p\eta_k\right)}{(n\lambda + \hat{V}p\eta_k)^2} \tag{30}$$

$$= \lim_{p\to\infty}\sum_{k=1}^{p}\frac{\frac{p^2}{n}\eta_k^2\hat{V}^2\epsilon_g + n^2\lambda^2\theta_k^{\star 2}\eta_k}{(n\lambda + \hat{V}p\eta_k)^2}, \tag{31}$$

thus

$$\epsilon_g = \lim_{p\to\infty}\frac{\frac{z^2}{n^2}\sum_{k=1}^{p}\frac{\theta_k^{\star 2}\eta_k}{\left(z\frac{1}{n}+\eta_k\right)^2} + \sigma^2}{1 - \frac{1}{n}\sum_{k=1}^{p}\frac{\eta_k^2}{(z\frac{1}{n}+\eta_k)^2}}. \tag{32}$$

Therefore, for the excess prediction error:

$$\epsilon_g - \sigma^2 = \lim_{p\to\infty}\frac{\frac{z^2}{n^2}\sum_{k=1}^{p}\frac{\theta_k^{\star 2}\eta_k}{\left(z\frac{1}{n}+\eta_k\right)^2} + \frac{\sigma^2}{n}\sum_{k=1}^{p}\frac{\eta_k^2}{(z\frac{1}{n}+\eta_k)^2}}{1 - \frac{1}{n}\sum_{k=1}^{p}\frac{\eta_k^2}{(z\frac{1}{n}+\eta_k)^2}}. \tag{33}$$

We now assume power-law form for the covariance spectrum and the teacher coordinates (8)

$$\eta_k = k^{-\alpha}, \qquad\qquad \theta_k^{\star 2}\eta_k = k^{-1-2r\alpha}, \tag{34}$$

Then equation (32) can be simplified to

$$\epsilon_g - \sigma^2 = \lim_{p\to\infty}\frac{\frac{z^2}{n^2}\sum_{k=1}^{p}\frac{k^{-1-2r\alpha}}{\left(z\frac{1}{n}+k^{-\alpha}\right)^2} + \frac{\sigma^2}{n}\sum_{k=1}^{p}\frac{k^{-2\alpha}}{(z\frac{1}{n}+k^{-\alpha})^2}}{1 - \frac{1}{n}\sum_{k=1}^{p}\frac{k^{-2\alpha}}{(z\frac{1}{n}+k^{-\alpha})^2}}, \tag{35}$$

which has a meaningful limit as $p \to \infty$ (with $n, \lambda$ kept fixed):

$$\epsilon_g - \sigma^2 = \frac{\sum_{k=1}^{\infty}\frac{k^{-1-2r\alpha}}{(1+nz^{-1}k^{-\alpha})^2} + \frac{\sigma^2 n}{z^2}\sum_{k=1}^{\infty}\frac{k^{-2\alpha}}{1+nz^{-1}k^{-\alpha})^2}}{1 - \frac{n}{z^2}\sum_{k=1}^{\infty}\frac{k^{-2\alpha}}{(1+nz^{-1}k^{-\alpha})^2}}. \tag{36}$$

Therefore, the excess prediction error suggestively decomposes into two terms, the first accounting for the variance due to sampling, while the second reflects the additional variance entailed by the label noise. Unlike a typical bias-variance decomposition, the effect of the bias (as manifested by the $\lambda$-dependent $z$ term) is subsumed in both terms. For simplicity, the first term in the numerator shall be referred to in the rest of the derivation as the *sample variance term*, and the second sum in the numerator as the *noise variance term*.

In the same limit, the equation defining $z$ (29) is amenable to being rewritten:

$$z = n\lambda + \frac{z}{n} \sum_{k=1}^{\infty} \frac{1}{1 + \frac{z}{n} k^{\alpha}}, \tag{37}$$

or, approximating the Riemann sum by an integral

$$z \approx n\lambda + \left(\frac{z}{n}\right)^{1-\frac{1}{\alpha}} \int_{\left(\frac{z}{n}\right)^{1/\alpha}}^{\infty} \frac{\mathrm{d}x}{1 + x^{\alpha}}. \tag{38}$$

### A.3 Infinite sample limit and the scaling of the generalisation error

Consider now the limit $n \gg 1$ with $\lambda$ scaling with n

$$\lambda \sim n^{-\ell}. \tag{39}$$

Note that the scalings of $z$ with respect to $n$ differ according to the regularisation $\lambda$, depending on which of the two terms on the right hand side of equation (37) dominates. If the first $n\lambda$ term dominates, then (37) simplifies to $z \approx n\lambda$. For this to be self-consistent, we must have $(z/n)^{1-\frac{1}{\alpha}} \approx \lambda^{1-\frac{1}{\alpha}} \ll n\lambda$, i.e. $n \gg \lambda^{-\frac{1}{\alpha}}$. In the converse case where the second term in (37) dominates, $z \sim n^{1-\alpha}$. For this to consistently hold, one needs $(z/n)^{1-\frac{1}{\alpha}} \approx n^{1-\alpha} \gg n\lambda$, i.e. $n \ll \lambda^{-\frac{1}{\alpha-1}}$. Depending on which term dominates in (37), two regime may be distinguished:

- In the *effectively non-regularized* $\ell > \alpha$ regime, $n \ll \lambda^{-\frac{1}{\alpha}}$ so $z \sim n^{1-\alpha}$. In this regime the regularization totally disappears from the analysis and KRR behaves just as if $\lambda = 0$.

- in the *effectively regularized* $\ell < \alpha$ regime, $n \gg \lambda^{-\frac{1}{\alpha}}$ regime, $z \approx n\lambda$.

### A.4 Effectively non-regularized regime

**Sample variance term:** As before, depending on $1 + 2r\alpha, \alpha$, it is sometimes possible to rewrite the sample variance term in integral form. If $r < 1$,

$$\sum_{k=1}^{\infty} \frac{k^{-1-2r\alpha}}{(1 + nz^{-1}k^{-\alpha})^2} \sim n^{-2r\alpha} \sum_{k=1}^{\infty} \frac{\left(\frac{k}{n}\right)^{-1-2r\alpha}}{(1 + \left(\frac{k}{n}\right)^{-\alpha})^2} \frac{1}{n} \sim n^{-2r\alpha} \int_0^{\infty} \frac{x^{-1+2(1-r)\alpha}}{(1 + x^{\alpha})^2} = \mathcal{O}(n^{-2r\alpha}). \tag{40}$$

If $r > 1$, it is no longer possible to write the Riemann sum as an integral, and

$$\sum_{k=1}^{\infty} \frac{k^{-1-2r\alpha}}{(1 + nz^{-1}k^{-\alpha})^2} = \sum_{k=1}^{n} \frac{k^{-1-2r\alpha}}{(1 + n^{\alpha}k^{-\alpha})^2} + n^{-2r\alpha} \sum_{k=n}^{\infty} \frac{\left(\frac{k}{n}\right)^{-1-2r\alpha}}{(1 + \left(\frac{k}{n}\right)^{-\alpha})^2} \frac{1}{n} = \mathcal{O}(n^{-2\alpha}). \tag{41}$$

**Noise variance term:** It is possible to similarly decompose the sum in the noise variance term to find

$$\frac{n\sigma^2}{z^2} \sum_{k=1}^{\infty} \frac{k^{-2\alpha}}{(1 + nz^{-1}k^{-\alpha})^2} = \mathcal{O}(\sigma^2). \tag{42}$$

From this, it follows that:

- for $n \ll \sigma^{-\frac{2}{2\alpha\min(r,1)}}$ the sample variance term dominates the numerator, and

$$\epsilon_g - \sigma^2 = \mathcal{O}(n^{-2\alpha\min(r,1)}) \tag{43}$$

- for $n \gg \sigma^{-\frac{2}{2\alpha\min(r,1)}}$ the noise variance term dominates the numerator, and determines the decay of the excess prediction error

$$\epsilon_g - \sigma^2 = \mathcal{O}(\sigma^2) \tag{44}$$

These two subregimes are amenable to being written in the more compact form (9):

$$\epsilon_g - \sigma^2 = \mathcal{O}\left(\max\left(\sigma^2, n^{-2\alpha\min(r,1)}\right)\right) \tag{45}$$

## A.5 Effectively regularized regime

**Sample variance term:** By the same token, in the second $\ell < \alpha$ regularized regime, provided $r < 1$, one can write the sample variance term as a Riemann sum (since $\lambda \sim n^{-\ell} = o(1)$):

$$\sum_{k=1}^{\infty} \frac{k^{-1-2r\alpha}}{(1+nz^{-1}k^{-\alpha})^2} \sim \lambda^{2r} \sum_{k=1}^{\infty} \frac{(k\lambda^{\frac{1}{\alpha}})^{-1+2(1-r)\alpha}}{\left((k\lambda^{\frac{1}{\alpha}})^{\alpha}+1\right)^2} \lambda^{\frac{1}{\alpha}} \sim \lambda^{2r} \int_0^{\infty} \frac{x^{-1+2(1-r)\alpha}}{(1+x^{\alpha})^2}$$

$$= \mathcal{O}(n^{-2\ell r}). \tag{46}$$

In the $r > 1$ case,

$$\sum_{k=1}^{\infty} \frac{k^{-1-2r\alpha}}{(1+nz^{-1}k^{-\alpha})^2} = \sum_{k=1}^{n^{\frac{\ell}{\alpha}}} \frac{k^{-1-2r\alpha}}{(1+\frac{1}{\lambda}k^{-\alpha})^2} + \lambda^{\frac{-2r\alpha}{\alpha}} \sum_{k=n^{\frac{\ell}{\alpha}}}^{\infty} \frac{(k\lambda^{\frac{1}{\alpha}})^{-1+2(1-r)\alpha}}{\left((k\lambda^{\frac{1}{\alpha}})^{\alpha}+1\right)^2} \lambda^{\frac{1}{\alpha}}. \tag{47}$$

Upper and lower bounds can be straightfowardly found for the first sum and the following equivalence established

$$\sum_{k=1}^{n^{\frac{\ell}{\alpha}}} \frac{k^{-1-2r\alpha}}{(1+\frac{1}{\lambda}k^{-\alpha})^2} \sim n^{-2\ell} \sum_{k=1}^{n^{\frac{\ell}{\alpha}}} k^{-1+2(1-r)\alpha} = \mathcal{O}(n^{-2\ell}), \tag{48}$$

while the second sum is a Riemann sum of order $\mathcal{O}(n^{\frac{(-2r\alpha)\ell}{\alpha}}) = o(n^{-2\ell})$. Therefore, the first sum in the numerator scales like

$$\sum_{k=1}^{\infty} \frac{k^{-1-2r\alpha}}{(1+nz^{-1}k^{-\alpha})^2} = \mathcal{O}\left(n^{-2\ell\min(r,1)}\right) \tag{49}$$

**Noise variance term:** The scaling of the noise variance term is found along similar lines to be

$$\frac{n\sigma^2}{z^2} \sum_{k=1}^{\infty} \frac{k^{-2\alpha}}{(1+nz^{-1}k^{-\alpha})^2} = \mathcal{O}(\sigma^2 n^{\frac{\ell-\alpha}{\alpha}}). \tag{50}$$

If the noise variance term decays faster in $n$, then the sample variance term always dominates (since $\sigma^2$ is at most $\mathcal{O}(1)$). This is the case when

$$0 < \ell < \frac{\alpha}{2\alpha\min(r,1)+1} \tag{51}$$

and then the generalization excess prediction error scales like

$$\epsilon_g - \sigma^2 = \mathcal{O}\left(n^{-2\ell\min(r,1)}\right). \tag{52}$$

In the case where $\alpha > \ell > \frac{\alpha}{2\alpha\min(r,1)+1}$ there exist two regimes depending on how $n$ compares with the noise strength

- if $n \ll \sigma^{\frac{2}{1-\frac{\ell}{\alpha}(1+2\alpha\min(r,1))}}$ the sample variance term dominates and we recover the noiseless case

$$\epsilon_g - \sigma^2 = \mathcal{O}\left(n^{-2\ell\min(r,1)}\right). \tag{53}$$

- if $n \gg \sigma^{\frac{2}{1-\frac{\ell}{\alpha}(1+2\alpha\min(r,1))}}$ the noise variance term dominates and

$$\epsilon_g - \sigma^2 = \mathcal{O}(\sigma^2 n^{\frac{\ell-\alpha}{\alpha}}). \tag{54}$$

All those regimes can be written more compactly as (10)

$$\epsilon_g - \sigma^2 = \mathcal{O}\left(\max\left(\sigma^2, n^{1-2\ell\min(r,1)-\frac{\ell}{\alpha}}\right) n^{\frac{\ell-\alpha}{\alpha}}\right). \tag{55}$$

**Case $\ell < 0$:** We give here for completeness the case in which the regularization grows with $n$. Then the sample variance term scales like

$$\sum_{k=1}^{\infty} \frac{k^{-1-2r\alpha}}{(1+nz^{-1}k^{-\alpha})^2} = \mathcal{O}(1). \tag{56}$$

To see this, use a lower and upper bound starting from $0 \le nz^{-1}k^{-\alpha} \sim n^\ell k^{-\alpha} \le 1$ for all $k \ge 1$ and all $n$. The noise variance term scales like

$$\frac{\sigma^2 n}{z^2} \sum_{k=1}^{\infty} \frac{k^{-2\alpha}}{(1+nz^{-1}k^{-\alpha})^2} \sim \sigma^2 n^{2\ell-1} = o(1), \tag{57}$$

meaning

$$\epsilon_g - \sigma^2 = \mathcal{O}(1) \tag{58}$$

## A.6 Continuity across the regularization crossover line

The $\ell = \alpha$ is actually comprised in the $\ell > 0$ case of the $\ell < \alpha$ regimes. On the $\ell = \alpha$ separation line, there is no discontinuity between the non-regularized exponents and the regularized exponents, since

$$\max \left( \sigma^2, n^{1-2\ell\min(r,1)-\frac{\ell}{\alpha}} \right) n^{\frac{\ell-\alpha}{\alpha}} \stackrel{\ell=\alpha}{=} \max \left( \sigma^2, n^{-2\alpha\min(1,r)} \right). \tag{59}$$

## A.7 Asymptotically optimal regularization

The derivation in subsections A.4 and A.5 effectively delimit the four regimes in Fig. 1: the effectively non-regularized noiseless green regime, the effectively regularized noiseless blue regime, the effectively non-regularized noisy red regime, and the effectively regularized noisy orange regime.

For any given $n$, we define the *asymptotically optimal* $\ell$ as the regularization decay yielding fastest decay of the excess prediction error. This corresponds to finding the $\ell$ with maximal excess error decay along a vertical line at abscissa $n$ in the phase diagram Fig. 1.

If $n \gg n_1^\star \approx \sigma^{-\frac{1}{\alpha\min(r,1)}}$ (effectively noisy regime), the noise-induced crossover line is crossed for

$$\ell_c \approx \left( 1 - 2\frac{\ln \sigma}{\ln n} \right) \frac{\alpha}{1+2\alpha\min(r,1)}. \tag{60}$$

The asymptotically optimal $\ell^\star$ is found as

$$\ell^\star = \operatorname*{argmax}_\ell \left( 2\ell\min(r,1)\mathbb{1}_{0<\ell<\ell_c} + \frac{\alpha-\ell}{\alpha}\mathbb{1}_{\ell_c<\ell<\alpha} + 0 \times \mathbb{1}_{\alpha<\ell} \right). \tag{61}$$

Since the argument of the argmax is an increasing function of $\ell$ on $(0, \ell_c)$ and a decreasing function on $(\ell_c, \infty)$ the maximum is found for $\ell^\star = \ell_c$. The corresponding decay for the excess error is

$$\max \left( 2\ell^\star\min(r,1), \frac{\alpha-\ell^\star}{\alpha} \right) = 2\ell^\star\min(r,1) \approx \frac{2\alpha\min(r,1)}{1+2\alpha\min(r,1)} \left( 1 - 2\frac{\ln \sigma}{\ln n} \right). \tag{62}$$

It is nonetheless ill-defined to talk about an aymptotically optimal rate for the regularization that continuously varies with $n$ when $n$ is comparable with $\sigma^{-2}$, since (62) means that the excess error is not even a power law in this region. An asymptotic statement can be however made. For $n \gg n_2^\star \approx \max(n_1^\star, \sigma^{-2})$,

$$\ell^\star \approx \frac{\alpha}{1+2\alpha\min(r,1)}, \tag{63}$$

and the excess error decays like (12)

$$\epsilon_g^\star - \sigma^2 = \mathcal{O} \left( n^{-\frac{2\alpha\min(r,1)}{1+2\alpha\min(r,1)}} \right). \tag{64}$$

For $n \ll \sigma^{-\frac{2}{2\alpha\min(r,1)}}$ (effectively noiseless regime), we have

$$\ell^\star = \operatorname*{argmax}_\ell \left( 2\ell\min(r,1)\mathbb{1}_{0<\ell<\alpha} + 2\alpha\min(r,1)\mathbb{1}_{\alpha<\ell} \right), \tag{65}$$

which means that any $\ell^\star \in (\alpha, \infty)$ is optimal (in particular, vanishing regularization is optimal), and we recover (11)

$$\epsilon_g^\star - \sigma^2 = \mathcal{O} \left( n^{-2\alpha\min(r,1)} \right). \tag{66}$$

## B A dictionary of notation in the literature

While the capacity and source conditions are assumed in almost all works concerned with the decay rates of Kernel methods, the actual notations for the capacity and source terms $\alpha, r$ greatly vary. We provide in this appendix a table summarizing notations for the references [8–15, 18]

| Reference | $\alpha$[8] | $r$[8] |
|---|---|---|
| [13] | $b$ | $\frac{a-1}{2b}$ |
| [12] | $\frac{\alpha_S}{d}$ | $\frac{1}{2}(\frac{\alpha_T}{\alpha_S} - d)$ |
| [9] | $\beta$ | $\frac{2\delta+\beta-1}{2\beta}$ |
| [10, 14] | $b$ | $\frac{c}{2}$ |
| [11, 15] | $\frac{1}{p}$ | $\frac{\beta}{2}$ |
| [18] | $b$ | $\beta$ |

Table 2: Dictionary between different notations previously used in the KRR literature.

## C Details on real data sets

### C.1 Feature map to diagonal covariance for real datasets

In the general case where the data $x$ is drawn from a generic distribution $\rho_x$, we remind the equations defining the feature map $\psi$ (3):

$$\psi(x) = \Sigma^{\frac{1}{2}}\phi(x) \tag{67}$$

$$\mathbb{E}_{x\sim\rho_x}\left[\phi(x)\phi(x)^T\right] = \mathbb{1}_p \tag{68}$$

$$\mathbb{E}_{x'\sim\rho_x}\left[K(x,x')\phi(x')\right] = \Sigma\phi(x) \tag{69}$$

In the of a real dataset $\mathcal{D} = \left\{x^\mu, y^\mu\right\}_{\mu=1}^{n_{\text{tot}}}$ from which both the train and test set are uniformly drawn, the distribution is then the empirical uniform distribution over $\mathcal{D}$,

$$\rho_x(\cdot) = \frac{1}{n_{\text{tot}}}\sum_{\mu=1}^{n_{\text{tot}}}\delta(\cdot - x^\mu). \tag{70}$$

Defining the Gram matrix $(K_{\mu\nu})_{\mu,\nu=1}^{n_{\text{tot}}} \overset{\text{def}}{=} (K(x^\mu, x^\nu))_{\mu,\nu=1}^{n_{\text{tot}}} \in \mathbb{R}^{n_{\text{tot}} \times n_{\text{tot}}}$, the equations defining the feature map (3) can be rewritten in the simpler matricial form

$$\psi = \phi\Sigma^{\frac{1}{2}}, \qquad \frac{1}{n_{\text{tot}}}\phi^T\phi = \mathbb{1}_{n_{\text{tot}}}, \qquad \frac{1}{n_{\text{tot}}}K\phi = \phi\Sigma \tag{71}$$

where $\phi, \psi, \lambda, K \in \mathbb{R}^{n_{\text{tot}} \times n_{\text{tot}}}$, and the feature space is of dimension $p = n_{\text{tot}}$, with the $\mu^{\text{th}}$ line of $\psi$ (resp. $\phi$) corresponding to $\psi(x^\mu)$ (resp. $\phi(x^\mu)$). To access the coordinates $\theta_k^\star$ in the basis of the features $\psi$, remember $\psi\theta^\star = y$, hence

$$\theta^\star = \frac{1}{n_{\text{tot}}}\Sigma^{-1}\psi^T y \tag{72}$$

### C.2 Estimation of source and capacity

The capacity and source terms $\alpha, r$ can be empirically estimated for the dataset $\mathcal{D}$ from the eigenvalues $\{\lambda_k\}_{k=1}^{n_{\text{tot}}}$ of the Gram matrix $K$ and the components $\{\theta_k^\star\}_{k=1}^{n_{\text{tot}}}$ of the teacher vector. Supposing decays like (8), the cumulative functions read:

$$\sum_{k'=k}^{n_{\text{tot}}}\lambda_{k'} \sim k^{1-\alpha}, \qquad \sum_{k'=k}^{n_{\text{tot}}}\lambda_{k'}\theta_{k'}^{\star 2} \sim k^{-2r\alpha}. \tag{73}$$

These functions are plotted in Fig. 7 and the terms $\alpha, r$ estimated therefrom. The use of the cumulative functions, rather than a direct estimation from the coordinates, allows the integration to smoothen out the curves and get a more consistent estimation. The values of $\alpha, r$ thereby measured are summarized in Table. 1. Note that the power-law form (8) and the assumption $p = \infty$ fail to hold for real data, and the series (73) have power-law form only on a range of indices $k$, before a sharp drop due to the finite dimensionality $n_{\text{tot}}$ of the feature space, see Fig. 7. The range of indices $k$ where the power-law form (8) seems to hold was qualitatively assessed, and linear regression run thereon to estimate $\alpha, r$. Since there is no clear objective way to determine the range the fit should be conducted on, the estimates slightly vary depending on the precise choice of the regression range, without however overly hurting the qualitative agreement with simulations Fig. 6 and 5.

### C.3 Details on simulations

We close this appendix by providing further detail on the simulations on real data (Figs. 6 and 5).

For each simulation at sample size $n$ a train set was created by subsampling $n$ samples from the total available dataset $\mathcal{D}$. To mitigate the effect of spurious correlations due to sampling a finite dataset, the whole dataset $\mathcal{D}$ has been used as a test set, following [25, 49]. A kernel ridge regressor was fitted on the train set with the help of the `scikit-learn KernelRidge` package [47]. For Fig. 6, the best regularization $\lambda$ was estimated using the `scikit-learn GridSearchCV` [47] default $5-$fold cross-validation procedure on the Grid $\lambda \in \{0\} \cup (10^{-10}, 10^5)$, with logarithmic step size $\delta \log \lambda = 0.026$. The excess test error was averaged over 10 independent samplings of the train set and noise realizations.

## D More crossovers

### D.1 Regularization-induced crossover

On top of the distinction between effectively noiseless regimes (green, blue regions in Fig. 1) and effectively noisy regimes (red, orange in Fig. 1), the four regimes can also be classified in effectively non-regularized (green, red) and effectively regularized (blue, orange), see also the discussion in Section 4. In Fig. 1, the non-regularized regions lie above the horizontal separation line $\ell = \alpha$, while the regularized ones lie below. In this appendix, we discuss a more generic setting for which this separation line ceases to be horizontal, thereby creating *a new crossover line*. Similarly to the noise-type crossover line discussed in the main text, a learning curve that crosses this regularization-induced crossover line transitions from an non-regularized regime (green, red) to a regularized one (blue, orange), characterized by differing decays for the excess error $\epsilon_g - \sigma^2$. In Fig. 8 and Fig. 9, the noise-type crossover line is depicted in red, while the regularization-type crossover line is in blue.

In this section we thus detail the more general setting

$$\lambda = \lambda_0 n^{-\ell}, \tag{74}$$

with, compared to the setup studies in the main text and Appendix A, an additional prefactor $\lambda_0$ to the regularization $\lambda$ that is allowed to be $\ll 1$. The particular case $\ell = 0, \lambda_0$ small, has been studied in [13], and has been shown to give rise to a crossover due to the regularization, on top on the one evidenced in the present work due to the noise.

- For small $n$, KRR focuses on fitting the spiked subspace comprising large variance dimensions, and satisfies the norm constraint introduced by the regularization on the lower importance subspace. This phenomenon can be loosely regarded as the bias version of the benign overfitting for noise variance ([38, 44]) : the bias induced by the loss of expressivity due to the norm constraint is effectively diluted over less important dimensions, thereby not impacting the generalization.

- For larger $n$, decreasing the excess error $\epsilon_g - \sigma^2$ requires a good KRR fit also on the subspace of lesser importance, and the regularization effect is felt. In a noiseless green-blue crossover, this results in a slower decay because of the bias introduced by regularizing. On a noisy red-orange crossover, the regularization conversely helps to mitigate the noise and enables the excess risk to decay again.

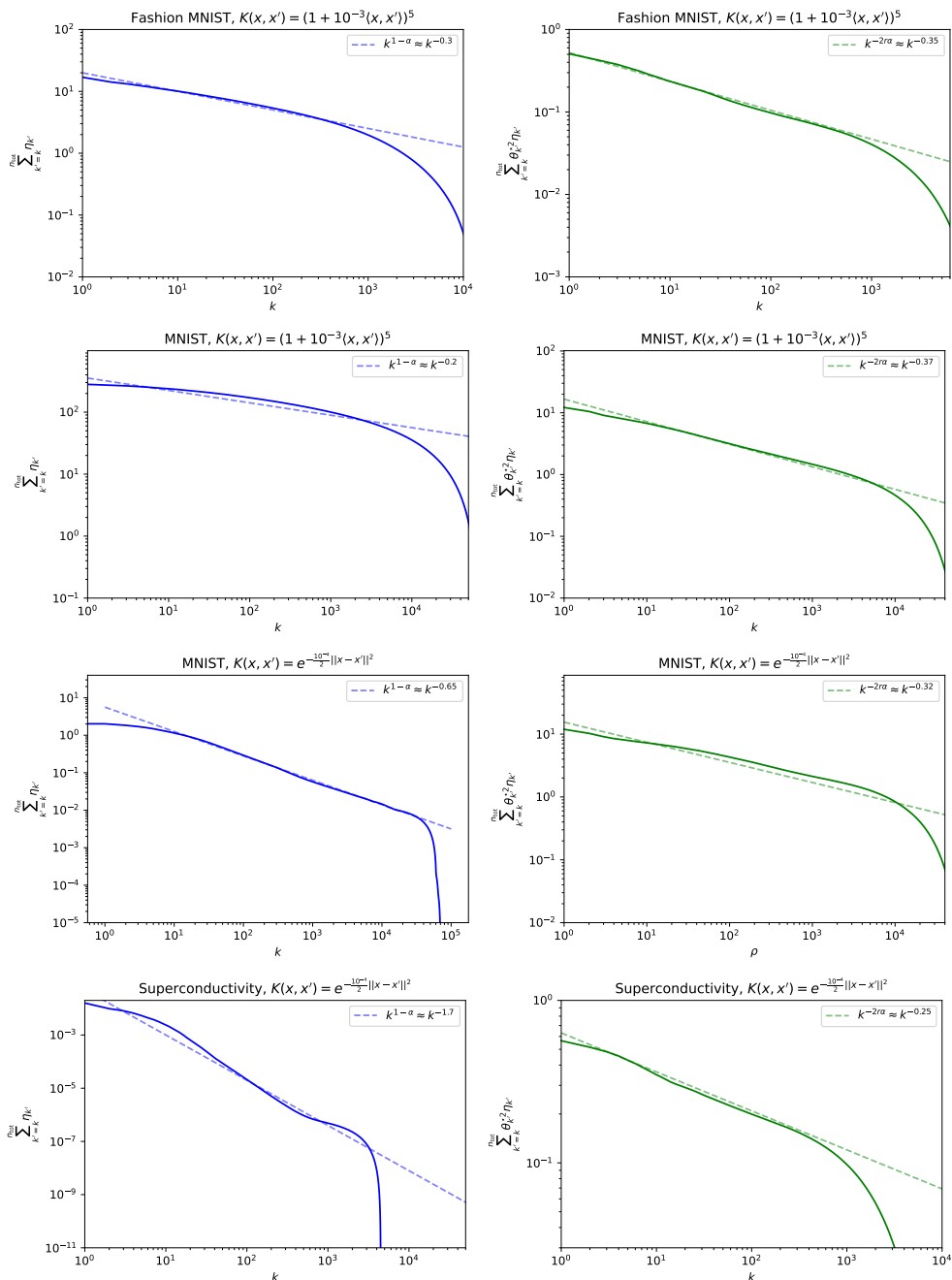

Figure 7: Measurement of empirical values for capacity and source $\alpha, r$ for real datasets (Fashion MNIST t-shirt and coat, MNIST) and RBF, polynomial kernels. Because the feature space is of finite dimension $n_{\text{tot}}$ all the curves exhibit a sharp drop at $n_{\text{tot}}$. A power-law was fitted on the functions (73) on a range of $k$ where these looked reasonnably like power-laws.

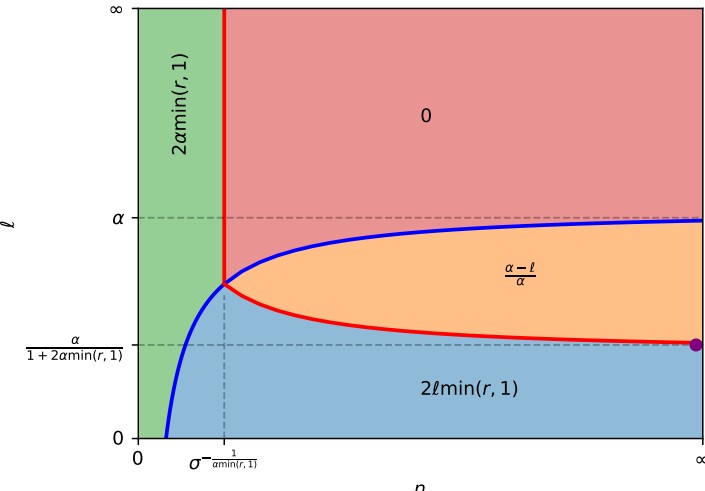

Figure 8: Phase diagram for $\lambda_0 \ll 1$ and $\sigma < \lambda_0^{\min(r,1)+\frac{1}{2\alpha}}$. As for Fig. 1. The solid red line corresponds to the noise-type crossover line, while the blue line indicates the regularization-type crossover line. Note that for low enough values of the regularization, the two crossover lines can be intercepted by a same horizontal line. This means that a double crossover is in theory observable (green-blue-orange), the first induce by regularization (see also [13]) and the second being noise-induced.

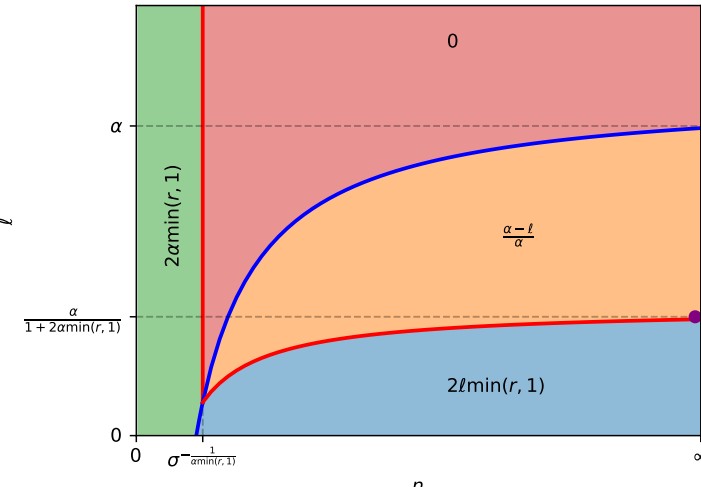

Figure 9: Phase diagram for $\lambda_0 \ll 1$ and $\sigma > \lambda_0^{\min(r,1)+\frac{1}{2\alpha}}$. As for Fig. 1, the asymptotically optimal decays $\ell^\star$ are indicated in solid red. The solid red line corresponds to the noise-type crossover line, while the blue line indicates the regularization-type crossover line. Note that for low enough values of the regularization, the blue crossover line can be intercepted by a horizontal line, alongside the red crossover line twice. Consequently a triple crossover is in theory observable (green-red-orange-blue), with two noise-induced and one regularization-induced.

## D.2 Outline of the computation

The derivation for the general case (74) follows very closely Appendix A.

- If $n \ll \lambda_0^{-\frac{1}{\alpha-\ell}}$ or $\ell > \alpha$, $n \ll \lambda^{-\frac{1}{\alpha}}$ so $z \sim n^{1-\alpha}$,

- If $n \gg \lambda_0^{-\frac{1}{\alpha-\ell}}$ and $\ell < \alpha$, $n \gg \lambda^{-\frac{1}{\alpha}}$ and $z \approx n\lambda$.

Note that the introduction of $\lambda_0 \ll 1$ means the limits between regularized and non-regularized regime are now involving both $n, \ell$ as opposed to just $\ell$ in Appendix A (see also Fig. 1). In the first $n \ll \lambda_0^{-\frac{1}{\alpha-\ell}}$ regime, the regularization effect is not sensed and the computation is identical to the $\lambda_0 = 1$ case in Appendix A. In the regularized $n \gg \lambda_0^{-\frac{1}{\alpha-\ell}}$, keeping track of the prefactors yields

$$\epsilon_g - \sigma^2 = \mathcal{O}\left(\lambda_0^{2\min(r,1)} n^{-2\ell\min(r,1)}\right) + \mathcal{O}\left(\sigma^2 n^{\frac{\ell-\alpha}{\alpha}} \lambda_0^{\frac{-1}{\alpha}}\right), \tag{75}$$

so the excess risk decays like

$$\epsilon_g - \sigma^2 = \mathcal{O}\left(\lambda_0^{2\min(r,1)} n^{\frac{\ell-\alpha}{\alpha}} \max\left(\left(\frac{\sigma}{\lambda_0^{\min(r,1)+\frac{1}{2\alpha}}}\right)^2, n^{-2\ell\min(r,1)+1-\frac{\ell}{\alpha}}\right)\right). \tag{76}$$

Depending on whether the maximum in (76) is always realized by one of its two arguments, or by one then the other as $n$ is varied, there may be a noise-induced crossover.

- if $\sigma < \lambda_0^{\min(r,1)+\frac{1}{2\alpha}}$ and $\ell \leq \frac{\alpha}{1+2\alpha\min(r,1)}$, the second argument of the maximum in (76) dominates for all $n \geq 1$ so no crossover is to be observed (see Fig. 8), and

$$\epsilon_g - \sigma^2 = \mathcal{O}\left(\lambda_0^{2\min(r,1)} n^{-2\ell\min(r,1)}\right). \tag{77}$$

- if $\sigma > \lambda_0^{\min(r,1)+\frac{1}{2\alpha}}$ and $\ell \geq \frac{\alpha}{1+2\alpha\min(r,1)}$, the first argument of the maximum in (76) dominates for all $n \geq 1$ so no crossover is to be observed (see Fig. 9), and

$$\epsilon_g - \sigma^2 = \mathcal{O}\left(\sigma^2 n^{\frac{\ell-\alpha}{\alpha}} \lambda_0^{\frac{-1}{\alpha}}\right). \tag{78}$$

- in any other case, a crossover between the decays (77) and (78) is observed, at a sample size

$$n_1^\star = \left(\frac{\sigma}{\lambda_0^{\min(r,1)+\frac{1}{2\alpha}}}\right)^{\frac{2}{1-\frac{\ell}{\alpha}(1+2\alpha\min(r,1))}}. \tag{79}$$

The crossover is from (77) to (78) if $\ell \geq \frac{\alpha}{1+2\alpha\min(r,1)}$ an in the other order if $\ell \leq \frac{\alpha}{1+2\alpha\min(r,1)}$.

The determination of the asymptotically optimal decays carries through as Appendix A, with the same conclusions. The four regimes and their respective limit, as well as the optimal $\ell$ at very large $n$ (purple point), are summarized in Figs. 8 and 9.

## D.3 Double and triple crossovers

We therefore recover the regularization induced crossover reported in [13] for the special case $\ell = 0, \sigma = 0$. It corresponds to the green-to-blue transition for the lowest $\ell$ in Fig. 8 9. We stress that such a mechanism is entirely due to the regularization, and hence happens *on top* of the noise-induced crossover studied in the present work. It is therefore possible in theory to observe both crossovers in succession.

We detail as an example a double green-to-blue-to-orange crossover (see blue curves in Fig.10). For small $n$ (non-regularized noiseless green regime), KRR fits the heavy dimensions. Both noise overfitting and bias are benign. As $n$ is increased, the blue regularization type crossover line in Fig. 8 is crossed and the regularized noiseless blue region entered. More of the less important dimensions need to be fitted well: bias is felt and entails a slower decay, but the noise overfitting is diluted over even less important dimensions and remains benign. As the red noise-type crossover line is passed into the regularized noisy orange region, the overfitting ceases to be benign and hurts the decay rate.

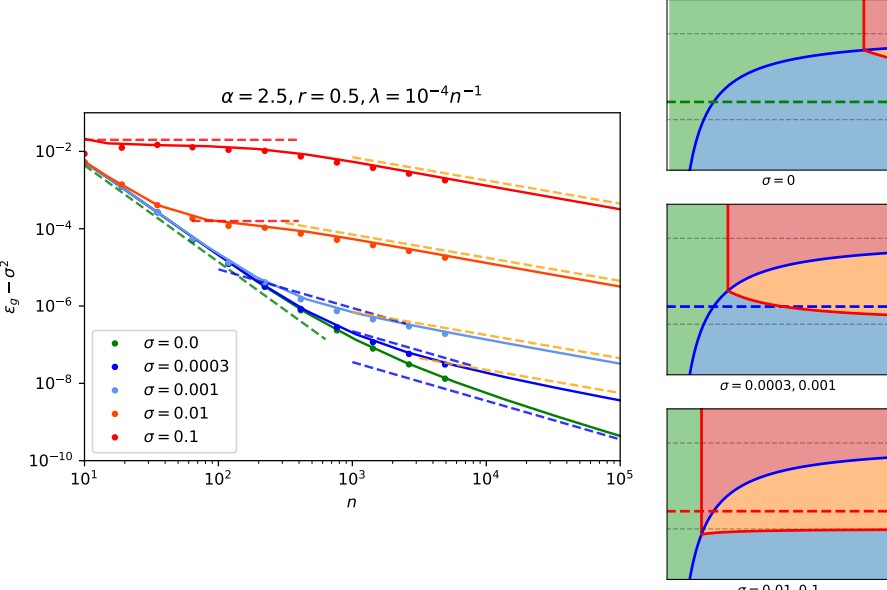

Figure 10: Excess risk learning curves for $\alpha = 2.5$, $r = 0.5$, $\lambda_0 = 10^{-4}$, $\ell = 1$. The noise level $\sigma$ is varied and the corresponding phase diagrams given on the right. For $\sigma = 0$ (green curve and top diagram on the right), a simple regularization-induced crossover (green-to-blue) is observed. For $\sigma = 3.10^{-4}$ and $\sigma = 10^{-3}$ (blue curves on the left, middle diagram on the right) a double crossover green-to-blue-to-orange is observed. The first is regularization induced, while the second is due to noise. For $\sigma = 10^{-2}$ and $\sigma = 10^{-1}$ (orange, red curves and bottom diagram), a double green-to-red-to-orange crossover is observed, respectively noise and regularization induced.

# E    Relation to worst-case bounds

In this section, we sketch informally how the blue and orange exponents (10) can also be derived from the worst case bounds [18, 19]. Note that the recovery from worst case bounds of the exponents (10), which were here derived in the Gaussian design setting, suggests that for these regimes the worst case exponents are also equal to the typical case exponents. We remind the reader that this has also already been shown to be the case for the asymptotically optimal lambda, see section 3, exponents (12) and [10, 14].

## E.1    Optimal rates for spectral algorithms with least-squares regression over Hilbert spaces [19]

To relate the notations employed in [19] to ours, the correspondances

$$\gamma \in ]0,1] = \frac{1}{\alpha}, \qquad \zeta \in [0,1] = r, \qquad \theta \in [0,1] = 1 - \ell, \qquad (80)$$

$$\mathcal{L} = \Sigma, \qquad f_H = f^\star, \qquad (81)$$

should be used, see also section B. With respect to their equation (18) defining the source condition, the setting considered in the present work corresponds to the special case $\phi(u) = u^\zeta$. Note that the assumption $\ell \leq 1$ is slightly more restrictive than those employed in this paper. The main result of [19] is their Theorem 4.2, which in our notations loosely translates to the following. With probability $1 - \delta$, there exist constants $\tilde{C}_1, \tilde{C}_2, \tilde{C}_3$ such that

$$(\epsilon_g - \sigma^2)^{\frac{1}{2}} \leq \left( \tilde{C}_1 n^{-\max(\frac{1}{2}, 1-r)} + \tilde{C}_2 n^{-\frac{1}{2}} \lambda^{\frac{-1}{2\alpha}} + \tilde{C}_3 \lambda^r \right) \ln \frac{6}{\delta} \left( \ln \frac{6}{\delta} + \frac{\max(\frac{1}{1-l}, \ln n)}{\alpha} \right), \quad (82)$$

i.e., explicating the scalings,

$$(\epsilon_g - \sigma^2)^{\frac{1}{2}} \leq \left( \tilde{C}_1 n^{-1 + \min(\frac{1}{2}, \min(r,1))} + \tilde{C}_2 n^{-\frac{1}{2}\frac{\alpha - \ell}{\alpha}} + \tilde{C}_3 n^{-\ell \min(r,1)} \right) \ln \frac{6}{\delta} \left( \ln \frac{6}{\delta} + \frac{\max(\frac{1}{1-l}, \ln n)}{\alpha} \right).$$
(83)

We replaced $\zeta$ by $\min(r,1)$ since [19] work under the assumption $\zeta = r \in [0,1]$ in order to make even clearer contact with the exponents in the present paper. Up to logarithmic corrections, one recognizes the blue ($\tilde{C}_3$ term in (83)) and orange ($\tilde{C}_2$ term in (83)) exponents, the effectively unregularized red and green regimes (9) being inaccessible in this setting because of the restriction $\ell \leq 1$. One can further show that the first $\tilde{C}_1$ term in (83) is always subdominant, since:

- if $r \geq \frac{1}{2}$, $n^{-\frac{1}{2} + \frac{\ell}{2\alpha}} \gg n^{-\frac{1}{2}}$ and the $\tilde{C}_2$ term dominates the $\tilde{C}_1$ term.
- if $r \leq \frac{1}{2}$, $n^{-1 + \min(1,r)} \ll n^{-\frac{1}{2}} \ll n^{-\ell \min(r,1)}$ since $\ell r \leq r \leq \frac{1}{2}$ and the $\tilde{C}_3$ term dominates $\tilde{C}_1$ term.

The relative competition between the $\tilde{C}_{2,3}$ contributions in (83) determine the blue to orange crossover (10), see discussion in section 3 of the main text. This suggests in particular that typical and worst case coincide within these two regimes.

### E.2 Kernel Truncated Randomized Ridge Regression: Optimal Rates and Low Noise Acceleration [18]

The notations can be mapped to those employed in the present paper in the following way:

$$\beta \in [0, \frac{1}{2}] = r, \qquad\qquad b \in [0,1] = \frac{1}{\alpha}. \tag{84}$$

Note that in [18], it is further assumed that the labels are bounded by a constant $Y$ while this only holds with high probability in our setting. The Theorem 3 in [18] then informally reads as: the error gap given by the KTR$^3$ algorithm [18] is approximately bounded by, for any $\epsilon_r, \epsilon_\alpha > 0$, for the power law ansatz (34):

$$\epsilon_g - \sigma^2 \leq \lambda^{2r - 2\epsilon_r} \frac{1}{2\alpha\epsilon_r} + \min\left[ \frac{4Y^2}{\alpha\epsilon_\alpha \lambda^{\frac{1}{\alpha} + \epsilon_\alpha} n} \min\left( \ln\left(1 + \frac{1}{\lambda}\right)^{1 - \frac{1}{\alpha} - \epsilon_\alpha}, \frac{\alpha}{1 + \alpha\epsilon_\alpha} \right), \right.$$
$$\left. \frac{\lambda^{2r - 2\epsilon_r - 1}}{2\alpha\epsilon_r n} + \frac{\sigma^2}{\lambda n} \right], \tag{85}$$

so in the particular setting $\lambda = n^{-\ell}$

$$\epsilon_g - \sigma^2 \leq n^{-2r\ell + 2\ell\epsilon_r} \frac{1}{2\alpha\epsilon_r} + \min\left[ \frac{4Y^2}{\alpha\epsilon_\alpha} n^{-\frac{\alpha - \ell}{\alpha} + \epsilon_\alpha \ell} \min\left( \ln\left(1 + n^\ell\right)^{1 - \frac{1}{\alpha} - \epsilon_\alpha}, \frac{\alpha}{1 + \alpha\epsilon_\alpha} \right), \right.$$
$$\left. \frac{n^{-2r\ell + 2\ell\epsilon_r + \ell - 1}}{2\alpha\epsilon_r} + \sigma^2 n^{-1 + \ell} \right]. \tag{86}$$

If $\sigma \neq 0$, the $\sigma^2 n^{-1 + \ell}$ term dominates in the second argument of the minimum and the minimum is realized by its first argument, leading to

$$\epsilon_g - \sigma^2 = \mathcal{O}(n^{-2\ell r}) + \mathcal{O}(n^{-\frac{\alpha - \ell}{\alpha}}) \tag{87}$$

namely the blue/orange crossover (10). If $\sigma = 0$ the bound is necessarily looser than $\mathcal{O}(n^{-2\ell r})$ which is coherent since in the noiseless setting only the blue exponent can be observed.