# OpenReview forum: "Generalization Error Rates in Kernel Regression: The Crossover from the Noiseless to Noisy Regime"
_NeurIPS.cc/2021/Conference — NeurIPS 2021 Poster_

### Official Review · Reviewer_Z2bD · 2021-07-15

**Rating:** 7
**Confidence:** 4

**Summary:**

The paper proposes an asymptotic analysis of the excess generalization error of kernel ridge regression under a gaussian design. In summary, the authors identify four regimes of functioning of kernel ridge regression as a function of the number of samples and the regularization parameter. These regimes allow unifying the theory of kernel ridge regression which had been performed previously in extreme cases (without noise, with noise but optimal regularization parameter).

**Limitations And Societal Impact:**

The limitations of the works are put in the work. For suggestions, the authors should consider trying to do some real-world experiments that are not played, clarify more specifically the link with exact (and not the order of magnitude as a function of sample size) asymptotic performed in RMT or at least what is the further insights with respect to reference [21] for example.

**Main Review:**

** Overall review **

1-Strong Points: The insights drawn are quite interesting, seem quite valid empirically on synthetic data, and allows for unifying the kernel ridge regression methods. The article is well written

2-Weak Points: The experiments in real data are not so convincing. It consists of self-added noise on the existing dataset (MNIST) which is far to be realistic. The question is how the theory can actually help to understand the excess generalization error of kernel ridge regression applied on existing datasets (which are already noisy by being collected manually, by humans, ...). Adding Gaussian noise is the same as creating synthetic data. The authors should also detail/motivate more the power-law decay assumption (should it model realistic features extracted from real dataset? Some real-world examples (figures of eigenvalues of the features co-variance) where this happens would also be interesting.)

** Detailed review **

1-Line 28: ... regression and are --> ... regression are

2-Line 59: a reference is missing

3-Line 96/97 Why is it important to assume this power-law decay (theoretical traceability, insights from experiments?). It should be good to precise.

4-In the paragraph Capacity and source coefficients, the authors should try to give a better interpretation, i.e., what does equation (7) mean in plain English?

5- The authors should try to make as possible their figure reproducible. For example, in Figure 3, it is impossible for a practitioner to obtain the figure (which kernel has been used for example, ...)

6-Line 177: for the the ---> for the

7-In figure 5, there is a reference for figures above and below.

8-The real data experiments on MNIST and Fashion MNIST are quite toyed examples and not really real data experiments. What is the motivation to add noise to the labels?

9- It would be better to make some connection between these predictions and the theoretical and exact large dimensional analysis of kernel ridge regression from Random Matrix Theory (RMT)? (How do they differ? Is the prediction of RMT be recovered?). There is a discussion in the last paragraph of 'Related work' but never discussed afterward in the main results.



**Time Spent Reviewing:**

6 hr

---

> ### Author Response · Authors · 2021-08-09
> **Reply to Reviewer Z2bD**
>
> We thank the referee for valuable pointers towards where the discussion of our paper should be improved.
>
> 1. *The experiments in real data are not so convincing. It consists of self-added noise on the existing dataset (MNIST) which is far to be realistic. The question is how the theory can actually help to understand the excess generalization error of kernel ridge regression applied on existing datasets (which are already noisy by being collected manually, by humans, ...).[...] The real data experiments on MNIST and Fashion MNIST are quite toyed examples and not really real data experiments. What is the motivation to add noise to the labels?*
>
> We completely agree with the referee that having validation of the theory for settings of real practical interest would be very nice. In effect, the noise in real data sets is often different from an additive Gaussian noise, and future research about deriving rates for more realistic noises would be valuable. Our motivation to nonetheless perform experiments for additive Gaussian noise is three-fold. First, it illustrates the theoretical predictions given in our work but also in many other previous works on the topic in a less idealistic setting than plain Gaussian inputs. Second, we observed that without adding noise most of the data sets we considered displayed the noiseless exponents. In order to illustrate the crossover phenomena discussed, we found it necessary to include additional noise. Finally, and perhaps most importantly, we wanted to connect the noiseless exponent with the classical - and abundant - literature on kernel ridge regression, where Gaussian additive noise is often considered.
>
>
> 2. *The authors should also detail/motivate more the power-law decay assumption (should it model realistic features extracted from real dataset? Some real-world examples (figures of eigenvalues of the features co-variance) where this happens would also be interesting.). Why is it important to assume this power-law decay (theoretical traceability, insights from experiments?). It should be good to precise.*
>
> Our primary reason to consider the power-law source/capacity conditions is that this is the most widely studied setting in the classical kernel regression literature, including many papers published at NeurIPS. It provides a well-posed theoretical problem encompassing many realistic settings. The main goal of this work is to clarify, unify and extend the apparently conflicting decay rates previously reported in the literature under this setting.
>
> However, we do agree with the referee that testing the power-law decay assumption (which most theoretical papers don't even do) is a really important question, and this is why we provided experiments with real data as illustration of our theoretical findings. The referee can find supporting plots in figure $7$ of the supplemental material, where it is apparent that a power-law fit is reasonable for the (simple) data sets considered in the present work. We stress, however, that many real data sets were found not to display power-law eigenvalue decay, see also the answer to referee JD5V. Building an analysis for error rates in cases where the power-law hypothesis fails to hold is an interesting future research avenue.
>
>
> **Detailed review** points 1., 2., 6., 7. We thank the referee for the list of suggestions and typos. We welcome all of them, and will correct in the revised version.  Points 3., and 8. are addressed in the answers above.
>
>
> 4. *In the paragraph Capacity and source coefficients, the authors should try to give a better interpretation, i.e., what does equation (7) mean in plain English?*
>
> We will make sure to motivate further the sourcce/capacity conditions in the revised manuscript. These names are taken from the statistical learning literature (e.g. [14]). The capacity $\alpha$ is a measure of the "effective dimension" - or equivalently the complexity - of the RKHS spanned by the kernel features. Looking at the Mercer decomposition $K(x,y) = \sum \eta_{i}\phi_{i}(x)\phi_{i}(y)$ it is clear that the faster $\eta_{i}$ decays (i.e. the bigger $\alpha$), less the small eigenvectors matter. For instance, if the RKHS is a Sobolev space the capacity condition is directly related to the smoothness of the functions in the RKHS. By the same token, the source $r$ measures the complexity of the target function: a large $r$ means that the target function is essentially lying on a "small" number of dimensions and is easier to learn.
>
> 5. *The authors should try to make as possible their figure reproducible. For example, in Figure 3, it is impossible for a practitioner to obtain the figure (which kernel has been used for example, ...)*
>
> The figures for synthetic datasets (Fig. 2,3,4) involve no specific kernel, as they are carried out by running ridge regression on Gaussian features, directly under the Gaussian design hypothesis, and with a co-variance being exactly the power law (8). In real settings, the kernel would determine the precise distribution of the features, and the exact covariance, but this dependency is omitted in Fig. $2,3,4$ which address directly the idealized Gaussian setting. We will clarify this in the captions.
>
> In any case, we will release clean python scripts for the full reproducibility of these experiments.
>
> 9. *It would be better to make some connection between these predictions and the theoretical and exact large dimensional analysis of kernel ridge regression from Random Matrix Theory (RMT)? .... clarify more specifically the link with exact (and not the order of magnitude as a function of sample size) asymptotic performed in RMT or at least what is the further insights with respect to reference [21] for example.*
>
> We departed our analysis using Ref. [23] but could have equivalently started from RMT formulas as in [21] or [39].  Many mathematically equivalent closed-form formulas for the prediction error exist in the literature for the very same ridge regression setting and we refer to a large numbers of these in lines 59-66 (including RMT, Gordon mini-max, leave-one-out, AMP...).
>
> While we make use of these works, we stress that none of them discusses the crossovers and the different regimes  summarized in Fig. 1, which is our main contribution with respect to e.g. [21].

---

> > ### Comment · Reviewer_Z2bD · 2021-08-25
> > **Rebuttal**
> >
> > Thank you for the detailed and accurate answers on the grey areas. I have re-evaluated the score of the review from 6 to 7. The justifications are as follows:
> >
> > 1-The article presents an interesting analysis, a missing point of view of asymptotic analysis of ridge regression of general interest to the ML community. The article is quite well written. The authors' response is sincere, honest, and clear.
> >
> > 2-While the authors underline the rather complex character of the noise in real data, for a practical impact of the analysis, it would be necessary to at least show that the theoretical analysis remains true or is robust for other noises than Gaussian ones (concentrated vectors for example,...) or to try to better model the noise in real data. The authors should highlight/comment on this point in the paper as a limitation.

---

> > > ### Author Response · Authors · 2021-08-27
> > > **Thank you for the feedback**
> > >
> > > We thank the referee for her/his positive feedback.
> > >
> > > We do agree that the assumption on the noise is a limitation of the current work, and will state this clearly in the revised manuscript.
> > >
> > > Our methods are conveniently not restricted to Gaussian noise, and we should be able to extend our analysis to other types of noise in future work. We thank the referee for the pointer to this interesting research direction.

---

### Official Review · Reviewer_rxhv · 2021-07-16

**Rating:** 6
**Confidence:** 4

**Summary:**

In this paper, the authors evaluate the excess error of Kernel Ridge Regression, in a unified manner by considering general settings for regularization and noise levels, as well as a power-law decay for the eigenvalues of feature covariance. The main results can be summarized in Figure 1, where one particularly observes the transitions between different regimes. Numerical results on real-world datasets such as MNIST, Fashion MNIST, and Superconductivity are also reported.

**Ethical Concerns:**

No.

**Limitations And Societal Impact:**

The authors adequately addressed the possible limitations of their approach. This work is mainly theoretical and algorithmic and I do not see any possible negative societal impact.


**Main Review:**

The paper is in general well written and easy to follow, the transition from the noiseless to noisy values reported here is, to the best of my knowledge, characterized for the first time in the general setting. I believe this work is of general interest to the NeurIPS community.

Detailed comments:
* Page 2, "Related work" section: the reference is missing for "the related Gaussian Process literature".

**Time Spent Reviewing:**

2

---

> ### Author Response · Authors · 2021-08-09
> **Reply to Reviewer rxhv**
>
> We thank the reviewer for his/her appreciation of our account of the different regimes in Fig. 1 that we also like very much.
>
> 1.  *Page 2, "Related work" section: the reference is missing for "the related Gaussian Process literature"*
>
> Thank you for the pointer, this was a compilation hitch, the forgotten reference is:
>
> *M. Kanagawa, P. Hennig, D. Sejdinovic, and B. K. Sriperumbudur. Gaussian processes and kernel methods: A review on connections and equivalences. arXiv:1805.08845v1 [stat.ML], 2018.*

---

### Official Review · Reviewer_Jd5V · 2021-07-17

**Rating:** 7
**Confidence:** 4

**Summary:**

This submission studies the asymptotic generalization error of kernel ridge regression (KRR) under Gaussian design with specific eigen-decay conditions. Major contributions include (i) specifying the decay rate of the generalization error based on the noise level and source / capacity conditions of the kernel / target function, (ii) identifying different regimes of learning, and (iii) deciding the optimal ridge regularization. Theoretical results are verified in various experiments.

**Main Review:**

## Strength

The excess risk of kernel regression has been extensively studied in the nonparametric regression literature. This paper provides a somewhat unifying perspective, which takes into account the noise level, different eigen-decay conditions and regularization strength.
The analysis builds upon recently derived precise asymptotics for ridge regression under general feature covariance and target function, and it is nice to see that these elusive asymptotic formulas can be used to provide quantitative understanding of classical machine learning models (instead of only generating good-looking double descent figures). To my knowledge, the different regimes of learning is an interesting contribution, and the matching empirical results on realistic datasets demonstrate the practical relevance of the theory.

## Weakness

1. The considered model is a bit idealized. I initially thought that the Gaussian design refers to the input features (as in [41] in the reference list), but instead it is directly assumed that the kernel features are Gaussian. It might be a good idea to comment on this gap, and whether the findings can be translated to the other setting via Taylor expansion on the kernel matrix.

2. Related to the previous point, due to the assumed Gaussian features, it seems that the generalization error of the studied estimator in the proportional limit has already been derived in prior works (e.g., [35][37]). The authors should comment on whether this is the case and the improvement over previous results.
For example, prior works in the proportional scaling usually assumed the population eigenvalues to be bounded away from 0, which cannot capture the power law decay. Would this be an advantage of the new analysis?

3. As the authors noted, the derivation of the excess risk rate is not completely rigorous due to the order of limits taken. While the empirical results are reassuring, would it be possible for certain terms that vanish in the proportional limit (and thus omitted in the asymptotic formulas) to reappear and dominate the rate when $p/n\to 0$?

**Additional Comments and Questions**

1. In the proportional limit, [35] showed that the optimal ridge regularization can be negative in the noiseless case under specific source conditions. Is it possible to observe similar phenomenon under the power-law eigen-decay setting? If not, what might be the reason for this difference?

2. Is there an intuition of why the power-law decay is a good fit for MNIST? The decaying eigenvalues for the capacity condition seems reasonable, but I don't really know why the source condition also follows a similar decay.

3. (minor) The reference [21] in Section 4 cannot handle the varying source condition due to the isotropic prior on the true coefficients. A more appropriate reference would be the general result of [35][37].

**Time Spent Reviewing:**

2-3 hours.

---

> ### Author Response · Authors · 2021-08-09
> **Reply to Reviewer Jd5V**
>
> We thank the reviewer for his/her appreciation and comments that help us see how to improve the presentation of our results.
>
> **Reply to points in "*Weakness*"**
>
> 1. *The considered model is a bit idealized. I initially thought that the Gaussian design refers to the input features (as in [41] in the reference list), but instead it is directly assumed that the kernel features are Gaussian. It might be a good idea to comment on this gap, and whether the findings can be translated to the other setting via Taylor expansion on the kernel matrix.*
>
> Gaussianity in the input space versus Gaussianity in the feature space: the referee is correct that we consider the latter. Encouraged by the presented numerical results on real data (that are not even Gaussian), we expect Gaussianity in the input space to translate into concentrated vectors  in the feature space under quite generic conditions, and thus that our Gaussian exponents hold broadly. This should be possible to establish analytically by putting together an extension of our results to concentrated vectors and the linearization used in [41]. For instance in [HL'20], it is proven for the specific case of random features that Gaussian data translate into equivalent Gaussian features.
>
> We agree with the referee extending the universality of the Gaussian design is an interesting direction to pursue and that it will require more work. See also answer to Reviewer 5R13 for a related discussion.
>
>
> [HL'20] *Hong Hu, Yue M. Lu, Universality Laws for High-Dimensional Learning with Random Features, arXiv abs/2009.07669, 2020*
>
> 2. *Related to the previous point, due to the assumed Gaussian features, it seems that the generalization error of the studied estimator in the proportional limit has already been derived in prior works (e.g., [35][37]). The authors should comment on whether this is the case and the improvement over previous results. For example, prior works in the proportional scaling usually assumed the population eigenvalues to be bounded away from 0, which cannot capture the power law decay. Would this be an advantage of the new analysis?*
>
> We acknowledge in the paper that the generalization error for the Gaussian features in the proportional limit has been derived in previous works, see e.g. lines 59-66, refs. [35, 37, 21, 36, 39, 23]. In particular, we build on the results of [23] that do not require the eigenvalues to be bounded away from zero and allow for the varying source condition. Removing these assumptions is thus not a contribution of our paper.
>
> The main contribution of our manuscript is to consider these previous work, explore the limit $n$ large, $p=\infty$ relevant to kernel regression, and unveil both theoretically (for the Gaussian design) and experimentally (for real data with added label noise) the crossover phenomena between noisy and noiseless regimes.
>
> We will state this more explicitly in the revised manuscript.
>
> 3. *[...] would it be possible for certain terms that vanish in the proportional limit (and thus omitted in the asymptotic formulas) to reappear and dominate the rate?*
>
> We believe this cannot happen. Our numerical checks reassure us in this perspective, as well as the experience with asymptotic results of the type [23] that numerically remains valid beyond the limit in which they are derived.
>
> Additionally, Theorem 5 in the last version of [30] provides a general non-asymptotic bound for the deviation between the finite-size risk and its asymptotic value in ridge regression. While this result is not readily applicable to our setting (due to our decaying $\lambda$), for to the reasons outlined above we believe it should be possible to derive a similar bound in our case, perhaps from the non-asymptotic bounds of [42]. But this would require a careful analysis.
>
> **Reply to points in "*Additional Comments and Questions*"**
>
> 1. *In the proportional limit, [35] showed that the optimal ridge regularization can be negative in the noiseless case under specific source conditions. Is it possible to observe similar phenomenon under the power-law eigen-decay setting? If not, what might be the reason for this difference?*
>
> We thank the reviewer for bringing up this interesting point.  Indeed, we have restricted ourselves to the classic regime with positive regularization, as our main motivation was to clarify the observed discrepancy of the error rates of kernel ridge regression in the literature [10,13,18,21].
>
>
> Following the referee's comment, we ran numerical simulations in a number of settings (various combination of $\alpha,r,\sigma$) by optimizing the excess error over $\lambda$, this time without the non-negativity constraint. We found that in the noiseless regime the optimal lambda can be indeed negative. Interestingly, however, the decay rate reported in the paper for the noiseless regime $2\alpha\mathrm{min}(1,r)$ seems to remain unchanged. We will investigate the theory for the $\lambda<0$ regime in future work, and thank the referee again for suggesting a new research avenue.
>
> 2. *Is there an intuition of why the power-law decay is a good fit for MNIST? The decaying eigenvalues for the capacity condition seems reasonable, but I don't really know why the source condition also follows a similar decay.*
>
> An insight is provided by equation (72) in Section C of the supplemental material. Since $\psi^T$ is a matrix whose rows decay according to a power law, and $y$ has components of order $1$, it follows that the teacher should approximately decay as a power law if the features approximately decay as such.
>
> However, we stress that there are also datasets and kernels for which a power-law decay is not a good fit for the eigenvalue distribution. For instance, that is the case for the Higgs dataset (comprising 28 features, with the target being $1$ if the event described corresponds to a Higgs boson and $0$ otherwise), for either the RBF or the polynomial kernel, or for MNIST with the Laplace kernel. Which dataset/kernel verify the power-law decay assumption (or fail to do so) is still a matter of current investigation.
>
>
> 3. *(minor) The reference [21] in Section 4 cannot handle the varying source condition due to the isotropic prior on the true coefficients. A more appropriate reference would be the general result of [35][37].*
>
> Thank you for this precision, we will adjust in the revision and refer to the pointed work more accurately.

---

> > ### Comment · Reviewer_Jd5V · 2021-08-23
> > **Additional Comments**
> >
> > Thank you for the detailed reply, which addressed some of my concerns; hence I have increased my score to 7. The observation that optimal negative regularization does not affect the generalization error rate is interesting -- I look forward to seeing further investigations.
> > One follow-up remark is that while I am aware of recent Gaussian equivalence results, I do not have a good intuition on what kind of setting (i.e., input distribution and teacher model) would give rise to the assumed power-law decay, at least for two-layer random features model in the proportional limit.
> > It might be a good idea to comment on this.

---

> > > ### Author Response · Authors · 2021-08-25
> > > **Thank you for the feedback**
> > >
> > > Thank you for your feedback and for the interesting remarks.
> > >
> > > About the last point: indeed for the random features in the proportional limit we do not get a power-law decay for the covariance spectrum (because the effective noise terms shifts the spectrum). This setting was not discussed in our paper, and we mentioned just as an example to answer the referee’s question about Gaussianity in the input space vs Gaussianity in the feature space. On the other hand, the power-law assumption is rather abundant in the kernel theory literature, which is the reason we adopted it. Looking at more generic cases thoroughly is an interesting topic for a future paper.

---

### Official Review · Reviewer_5R13 · 2021-07-17

**Rating:** 7
**Confidence:** 3

**Summary:**

The paper considers kernel ridge regression under the Gaussian design and derives decay rates of the excess generalization error under capacity and source conditions. The analysis reveals interesting phase transition phenomena in the noiseless and noisy regimes.

**Limitations And Societal Impact:**

Yes.

**Main Review:**

The presented decays rates of the excess generalization error of kernel ridge regression (KRR) under Gaussian design and the observed phase transition phenomena are of interest. The first result in (9) is the excess error of effectively non-regularized KRR, which transitions from a fast decay to a plateau with no decay as the sample size $n$ increases. The second result in (10) is the excess error of effectively regularized KRR, which transitions from a fast decay to a slower decay as $n$ increases. The results in (11) and (12) are the excess error of optimally regularized KRR, which transitions from a fast decay (optimal rate in noiseless regime) to a slower decay (optimal rate in noisy regime) as $n$ increases. The numerical experiments also verify their theoretical claims about the crossover from the noiseless to noisy regime. Overall, the paper is well written and the results are theoretically sound. It sheds light on how the decay rates of the excess error of KRR depend on the regularization decay strength, capacity, source, and noise variance. The Gaussian design is a rather strong condition for KRR, is it possible to remove this assumption in the future study?

After Rebuttal

I thank the authors for addressing my concerns.



**Time Spent Reviewing:**

3

---

> ### Author Response · Authors · 2021-08-09
> **Reply to Reviewer 5R13**
>
> We thank the referee for her/his appreciation of our work.
>
> 1. *The Gaussian design is a rather strong condition for KRR, is it possible to remove this assumption in the future study?*
>
> The difference between the Gaussian and generic  or even worst case design is indeed a very interesting question that partly motivated our work. One of our main results is showing that the optimally regularized Gaussian exponents (purple point in Fig. 1) agrees with the worst case results known in the literature.
>
> After submitting our paper we also noticed that two of the Gaussian exponents presented in our work (the "blue" and "orange" ones) follow (after some algebra) from the worst-case bounds of reference [LRRC 18] (see below). We will add the corresponding derivation linking to their bounds in the revised version.
>
> The comparison between the Gaussian "red" and "green" exponents and the corresponding worst-case results e.g. [10,18] remains unclear to us and we certainly want to address this in future work.
>
> Irrespective of the comparison with the worst case exponents, we do expect that the Gaussian assumption can be relaxed in all the considered regimes to sub-gaussian as in [42], or even to any concentrated distribution [T'95, CLC'2018], so that the results should be universal for a large class of distributions.  We will add a comment in the discussion in the revised version.
>
> For further discussion, please see also the answer to referee Jd5V.
>
> [LRRC'18] *Junhong Lin, Alessandro Rudi, L. Rosasco, and V. Cevher. Optimal rates for spectral algorithms with least-squares regression over Hilbert spaces. Applied and Computational Harmonic Analysis, 48:868–890, 2018*
>
> [T'95]: *Talagrand Michel, Concentration of measure and isoperimetric inequalities in product spaces,  Publications Mathématiques de l'IHÉS, Tome 81 (1995) , pp. 73-205., 1995*
>
> [CLC'2018]: *Cosme Louart, Zhenyu Liao, Romain Couillet, A random matrix approach to neural networks. The Annals of Applied Probability, 28(2) 1190-1248 April 2018.*

---

### Decision · Program_Chairs · 2021-09-27

**Decision:**

Accept (Poster)

**Comment:**

This paper studies asymptotic analysis of the excess generalization error of kernel ridge regression under a gaussian design. The authors identify four regimes of functioning of KRR as a function of the number of samples and the regularization parameter. These regimes allow unifying the theory of kernel ridge regression which had been performed previously in extreme cases. Numerical results on real-world datasets are also reported.

All the reviewers agree that the paper is well written, easy to follow and brings interesting theoretical insights. It’s pointed out that the paper provides `a missing point of view of asymptotic analysis of ridge regression of general interest to the ML` (reviewer `Z2bD`), `transition from the noiseless to noisy values reported here is, to the best of my knowledge, characterized for the first time in the general setting` (reviewer `rxhv`), `sheds light on how the decay rates of the excess error of KRR depend on the regularization decay strength, capacity, source, and noise variance` (reviewer `5R13`) and `different regimes of learning is an interesting contribution, and the matching empirical results on realistic datasets demonstrate the practical relevance of the theory` (reviewer `Jd5V`). Some of the limitations and concerns raised are addressed during the discussion period (authors promised to clarify limitations regarding the noise assumption and their potential robustness in the revision). In the end, all reviewers recommended for acceptance  (3 accepts, 1 weak accept). The AC believes this is a solid work that would be of interest to the broad NeurIPS audience.